# OpenHype: Hyperbolic Embeddings for Hierarchical Open-Vocabulary Radiance Fields

**Lisa Weijler**[*]
TU Wien

**Sebastian Koch**
Ulm University

**Fabio Poiesi**
Fondazione Bruno Kessler

**Timo Ropinski**
Ulm University

**Pedro Hermosilla**
TU Wien

## Abstract

Modeling the inherent hierarchical structure of 3D objects and 3D scenes is highly desirable, as it enables a more holistic understanding of environments for autonomous agents. Accomplishing this with implicit representations, such as Neural Radiance Fields, remains an unexplored challenge. Existing methods that explicitly model hierarchical structures often face significant limitations: they either require multiple rendering passes to capture embeddings at different levels of granularity, significantly increasing inference time, or rely on predefined, closed-set discrete hierarchies that generalize poorly to the diverse and nuanced structures encountered by agents in the real world. To address these challenges, we propose OpenHype, a novel approach that represents scene hierarchies using a continuous hyperbolic latent space. By leveraging the properties of hyperbolic geometry, OpenHype naturally encodes multi-scale relationships and enables smooth traversal of hierarchies through geodesic paths in latent space. Our method outperforms state-of-the-art approaches on standard benchmarks, demonstrating superior efficiency and adaptability in 3D scene understanding.

## 1   Introduction

Understanding 3D scenes requires capturing their inherent semantic hierarchical structure, as objects and their relationships are naturally organized in a multi-scale fashion [28]. For example, objects are composed of multiple parts, such as the arms and legs of a chair, and can also be semantically grouped at a higher level, e.g. a fridge and cabinets within a kitchen, or a couch and TV within a living room. Preserving this hierarchical organization is critical for a range of applications, including semantic segmentation, scene reconstruction, and object detection.

Natural language provides a human-readable interface for interacting with computational representations of 3D scenes. However, enabling such an interaction requires more than the mere processing of language input aligned with visual representations; it also requires the ability to model semantics at multiple scales and to reason about long-tail and abstract concepts. This requirement is particularly pronounced in the task of 3D open-vocabulary segmentation from Neural Radiance Fields (NeRF) [29], where a holistic understanding of the scene is essential for handling queries of varying granularity. Despite its importance, most existing methods overlook the hierarchical nature of 3D scenes. Instead, they treat 3D scenes as flat collections of independent elements and process the entire scene uniformly, generating independent embeddings for each spatial position [19, 15, 25, 7], often resulting in suboptimal performance when hierarchical understanding is required [1].

---

[*]lweijler@cvl.tuwien.ac.at, lisaweijler.github.io

39th Conference on Neural Information Processing Systems (NeurIPS 2025).

One promising direction for improving 3D scene understanding is the incorporation of hierarchical representations [15, 35, 1]. Introducing such inductive biases during training has been shown to enhance performance on tasks like semantic segmentation and object localization [35, 1]. However, existing NeRF-based approaches often require multiple rendering passes to obtain embeddings at different resolutions [15, 35], or depend on predefined, discrete hierarchical levels [35, 1]. These constraints limit the applicability in complex real-world scenarios, where hierarchical structures are not neatly constrained by fixed or discrete resolutions.

To overcome these challenges, we propose OpenHype, a novel method for open-vocabulary segmentation on NeRFs that leverages hyperbolic spaces to embed hierarchical structures in a continuous manner. Thanks to the exponential expansion rate of hyperbolic spaces [34], we are able to naturally embed hierarchies continuously, and later navigate such hierarchies by simply traversing geodesic paths within the latent space. OpenHype naturally accommodates varying levels of abstraction without the need for explicit discrete hierarchies or additional rendering passes, offering a more flexible and efficient framework for 3D scene understanding. We evaluate OpenHype both quantitatively and qualitatively on existing benchmarks, where we consistently outperform state-of-the-art methods.

## 2 Related Work

**Open-vocabulary 3D scene understanding.** Open-vocabulary scene understanding has gained considerable attention in recent years, shifting the paradigm in object detection and segmentation from a closed set of object classes to an open set [8, 23, 45, 47]. In 2023, this concept was popularized in the field of 3D scene understanding with OpenScene [33]. OpenScene jointly embeds 3D point features with image pixels and text to enable zero-shot 3D scene understanding. Follow-up works [2, 19, 15, 7, 25, 35] extend this idea to other 3D representations, such as Neural Radiance Fields (NeRF) [29]. Distilled Feature Fields (DFFs) [19] addresses the challenge of semantic scene decomposition in NeRF, enabling query-based local editing of 3D scenes. Later, Language Embedded Radiance Fields (LERF) [15] integrated language embeddings from foundation models like CLIP [37] into NeRF, enabling open-ended language queries in 3D. LERF learns a dense, multi-scale language field within NeRF by volume-rendering CLIP embeddings along training rays, and incorporates DINO features [3] for language regularization. Similar to LERF, OpenNeRF [7] encodes CLIP features within NeRF. However, OpenNeRF utilizes pixel-wise CLIP features instead of global CLIP features, resulting in a simpler architecture that eliminates the need for DINO regularization. Unlike NeRF-based methods, LangSplat [35] leverages the 3D Gaussian Splatting (3DGS) [14] representation to encode language features distilled from CLIP into each 3D Gaussian, enabling efficient rendering and open-vocabulary 3D understanding. Concurrently, Feature 3DGS [50] extended 3DGS toward more general multi-modal feature learning. While LangSplat employs an autoencoder to compress high-dimensional CLIP features before integrating them into the Gaussian representation, Feature 3DGS introduces a lightweight upsampling network that enables training low-dimensional feature fields jointly with radiance information. In contrast, Semantic Gaussians [9] bypass explicit distillation during training by projecting 2D semantic features onto 3D Gaussians through pixel–Gaussian correspondence and multi-view feature aggregation. While it demonstrates fine-grained capabilities such as part segmentation through features inherited from 2D models like VLPart, its semantics remain per-Gaussian and non-hierarchical, lacking explicit multi-level organization as enabled by our approach. Recently, several methods also extend 3DGS approaches with feature quantization methods: Shi et al. [40], Wu et al. [44]. In this paper, we extend these works by directly learning continuous hierarchical embeddings within NeRF.

**Hierarchical 3D scene understanding.** Hierarchical representations have long been central to how neural networks interpret complex data, mirroring the multi-level abstraction observed in human perception. In computer vision, convolutional neural networks (CNNs) naturally build hierarchical features, progressing from low-level edges to high-level object parts and categories [21, 38, 11]. However, despite their success on downstream tasks, those hierarchies are fixed, defined by the network architecture, and treated as a black box, making these models difficult to interpret. In 3D scene understanding, hierarchical representations have also gained attention, and some approaches do incorporate such inductive biases. Search3D [42], for example, segments 3D scene entities, such as object instances or object parts, from point clouds, based on arbitrary textual queries. Thus, it can construct hierarchical open-vocabulary 3D scene representations with two pre-defined levels, enabling 3D search at two semantic granularity levels. DeCompositional Consensus [31] also incorporated

hierarchical understanding in 3D point clouds, by maximizing agreement between segmentation hypotheses and their decomposed parts with a part segmentation network. However, those methods depend on large data sets for training and cannot be directly applied to NeRF representations.

**Hierarchical representations in radiance fields.** In NeRF, several methods have been proposed to incorporate hierarchical representations during learning [16, 15, 35, 1, 44]. GARField [16] decomposes 3D scenes into a hierarchy of semantically meaningful groups by optimizing a scale-conditioned 3D affinity feature field, allowing a point in space to belong to different groups at different scales. This field is optimized using 2D masks obtained from a Segment Anything (SAM) [17] foundation model. More related to our work, LERF [15] incorporates hierarchical language embeddings into NeRF by conditioning the 3D feature field with a scale parameter. Supervision of such a hierarchical model was achieved by computing CLIP embeddings with bounding boxes of multiple sizes. During inference, multiple rendering passes are required to obtain multiple feature maps. LangSplat [35] leverages SAM to learn hierarchical semantics at three different scales, reducing the extensive language field queries to three. Concurrent to our work and building on LangSplat, Hi-LSplat [49] further refines hierarchical understanding through a multi-scale latent-space disentanglement strategy. Hi-LSplat constructs a hierarchical 3D semantic tree via clustering and contrastive learning, yet still relies on the three predefined scale levels of SAM. Recently, N2F2 [1] has proposed to avoid relying on scale-conditioned language fields and instead learn a single feature field that contains hierarchical information. N2F2 learns a single feature using Matryoshka Representation Learning [22] that can be later processed to derive the language embedding at the desired granularity. However, the number of hierarchical levels is fixed and equal to the number of dimensions of the embedding, which might lead to insufficient representation power if the number of dimensions is too small or too large. The paper also proposes to learn a weighted sum of the different hierarchical embeddings from general concepts to reduce the number of queries to one; however, as our experiments will show, this results in subpar understanding compared to OpenHype.

In contrast to previous approaches, we propose to learn a single feature field in hyperbolic space that naturally encodes the hierarchy of the scene, allowing for a continuous navigation of such hierarchy by interpolating the embedding along a geodesic path. Furthermore, contrary to LERF [15], LangSplat [35], and Hi-LSplat [49], our method only requires a single rendering pass, and our continuous hierarchical embeddings in hyperbolic space lift the constraints of discrete hierarchical levels inherent to N2F2 [1].

## 3 Preliminaries

In this section, we briefly introduce the key concepts of hyperbolic geometry on which OpenHype relies. Hyperbolic spaces are Riemannian manifolds with constant negative curvature, which causes the volume of a ball to grow exponentially with its radius. By contrast, Euclidean space has zero curvature and therefore exhibits only polynomial volume growth. Intuitively, hyperbolic spaces provide "more room" than Euclidean ones, which makes them well-suited for embedding trees with arbitrarily small distortion [39]. Several isometric models exist for representing hyperbolic spaces [34], among which the Poincaré ball and the Lorentz model are the most widely used for hierarchical embeddings. In OpenHype, we use the Lorentz model $\mathbb{L}^n$, also known as the hyperboloid, due to its numerical stability, computational efficiency during optimization [30, 32], and a more stable distance function [34, 32], which we use in our hierarchical loss.

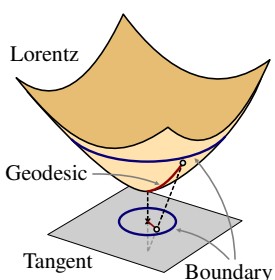

**Geodesics.** The shortest path between two points in $\mathbb{L}^n$ is called a *geodesic*. These paths naturally curve inward, rather than staying straight as in Euclidean geometry, due to the curvature of the space. In our approach, we use geodesics between the origin of the hyperboloid and feature embeddings to traverse object hierarchies. Distances on the Hyperboloid are measured using the Lorentzian geodesic distance $d_{\mathbb{L}}(\boldsymbol{x}, \boldsymbol{y})$.

**Exponential and logarithmic maps.** Another important concept of hyperbolic geometry is exp. and log. maps, which allow us to move from the tangent space, at a point $\boldsymbol{x} \in \mathbb{L}^n$, onto the Hyperboloid $\mathbb{L}^n$ and back, respectively. Since tangent spaces are Euclidean spaces, the outputs of Euclidean

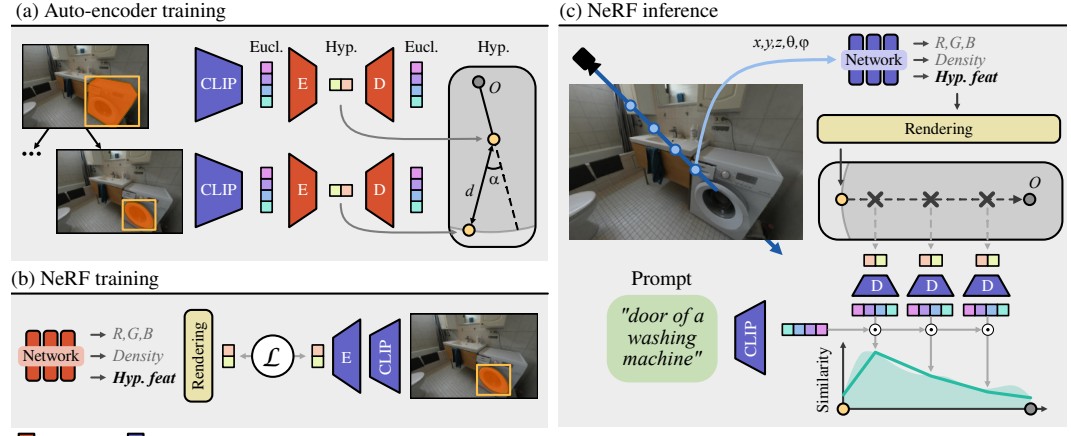

**(a) Auto-encoder training**

**(b) NeRF training**

**(c) NeRF inference**

■ Trainable ■ Frozen

Figure 1: **Method overview figure. (a)** Our auto-encoder transforms a hierarchy of language-aligned features into hyperbolic space. **(b)** Then, we supervise a NeRF model to predict the hierarchical hyperbolic features together with color and density. **(c)** Lastly, during inference, we can traverse in a continuous manner the scene hierarchy by following a geodesic path in hyperbolic space and perform open-vocabulary queries for a given text prompt.

network layers can be interpreted as vectors in a tangent space of the Lorentz model. These vectors can then be easily mapped onto the hyperboloid, enabling the learning of embeddings in hyperbolic space. Similar to Desai et al. [6], in OpenHype, we only consider the tangent space of the Hyperboloid's origin $O = [\mathbf{0}, \sqrt{1/c}]$. We refer the reader to the supplementary material for additional formal definitions and equations.

## 4 Method

Our hierarchical framework OpenHype takes as input a collection of images representing the scene with the associated camera parameters, and trains a NeRF model to produce 3D consistent hierarchical semantic features Fig. 1. However, contrary to existing approaches, OpenHype features are hyperbolic embeddings and thus allow for continuous hierarchy traversal along geodesic paths. As illustrated in Fig. 1 (a), OpenHype utilizes an auto-encoder that transforms language-aligned features, extracted from the scene images, to the hyperbolic space, which encodes the hierarchical structure of the scene (Sec. 4.1). Our NeRF model is supervised with hyperbolic features (Sec. 4.2 and Fig. 1 (b)), which then, during inference, can be queried in a continuous manner to traverse multiple levels of the scene hierarchy to acquire multi-scale responses for a given input text prompt (Sec. 4.3 and Fig. 1 (c)).

### 4.1 Hyperbolic auto-encoder

Our auto-encoder takes as input language-aligned $D$-dimensional features, $\mathcal{F} \in \mathbb{R}^{N \times D}$, of $N$ different objects (parts) present in the scene, $\mathcal{O}$, and the hierarchical relationship between them, $\mathcal{R} \in \mathbb{R}^{N \times N}$, where $R_{ij} = 1$ if object $i$ is the parent of object $j$ and $R_{ij} = 0$ otherwise. The encoder processes $\mathcal{F}$, mapping it to a hyperbolic latent $H$-dimensional representation, $\mathcal{H} \in \mathbb{R}^{N \times H}$. In this hyperbolic space, object $o_j$, with associated feature $h_j$, is an ancestor of object $o_i$, with associated feature $h_i$, if $d_{\mathbb{L}}(h_j, O) < d_{\mathbb{L}}(h_i, O)$ and $\alpha(h_j, h_i) \approx 0$, i.e., $h_j$ is closer to the origin than $h_i$, and $h_j$ lies close to the geodesic path connecting $h_i$ and the origin $O$. This results in leaf nodes (e.g. parts of objects) embedded close to the boundary[1] of the hyperbolic space and objects high in the hierarchy embedded close to the origin $O$. At the same time, elements of a branch in the hierarchy are embedded close to the same geodesic path connecting $O$ and a point on the boundary of the hyperbolic space. The decoder then takes the hyperbolic features and reconstructs them back to the original language features living in Euclidean space. Fig. 1 (a) illustrates this process.

---

[1]Note that this isn't a boundary of the space in a mathematical sense but rather a "boundary orbit" that we introduced by limiting the maximum geodesic distance to the origin to ensure numerical stability.

**Loss.** In order to enforce such a structure in the hyperbolic space, we employ contrastive learning to train the auto-encoder. From a batch of features processed, we compute the following loss: $\mathcal{L} = \mathcal{L}_d + \mathcal{L}_a + \mathcal{L}_r$. $\mathcal{L}_d$ is the standard contrastive learning loss [4, 37, 6], but minimizing hyperbolic distances instead of maximizing cosine similarities. $\mathcal{L}_a$ uses the same contrastive approach as $\mathcal{L}_d$ but applied to the exterior angle, $\pi - \angle Oh_j h_i = \alpha(h_j, h_i)$. Fig. 1 (a) illustrates the exterior angle. Lastly, $\mathcal{L}_r$ is the reconstruction loss of the decoded language-aligned features, defined as the Mean Squared Error (MSE) between the reconstructed feature $f_i'$ and the input feature $f_i$.

**Hierarchical data creation.** In order to train the auto-encoder and later supervise the NeRF model, we need to extract the set of objects $\mathcal{O}$, associated features $\mathcal{F}$, and the hierarchical relationships $\mathcal{R}$ from the collection of input images $\mathcal{I}$. We leverage a recent foundation model for general object segmentation, Semantic SAM [24], to generate a set of object masks $\mathcal{M}$ from the input images. For each mask $m_i$, we extract the language-aligned features $f_i$ associated with object $o_i$ by cropping the image and processing it with CLIP [37]. The cropped image only contains pixels within the bounding box of the object mask $m_i$. Moreover, for each image, we compute a hierarchy representing the relationship between objects present in the image. First, the largest masks, those that are not contained in any other mask, are directly associated with the root node, defining the level $l_1$ of the hierarchy. Then, we iterate over all the remaining masks in the image and select those that are only contained within the masks of level $l_1$. These new masks are associated with the corresponding mask of $l_1$, defining the new level $l_2$ in the hierarchy. The process is repeated until there are no masks left in the image. Note that, even if the same object appears in multiple images, each image will have its own instance of the same object and will be part of a different hierarchy. However, due to the generalization power of visual-language foundation models such as CLIP, the resulting features will be similar. As we will show later in our experiments, during the auto-encoding training process, these features will be transformed to the same region of the hyperbolic space, consolidating the hierarchies of multiple images and creating a unique hierarchical structure for the scene.

## 4.2 3D lifting

A radiance field defines a function that maps a 3D point $\mathbf{x} \in \mathbb{R}^3$ and a viewing direction $\mathbf{d} \in \mathbb{S}^2$ to both a color value $\mathbf{c} \in [0,1]^3$ and a volume density $\sigma \in [0, \infty)$. Mildenhall et al. [29] introduced NeRF, which models this function implicitly using an Multi Layer Perceptron (MLP). The network, parameterized by $\theta$, is trained to regress the radiance values from a set of multi-view images of a scene. Formally, the network is defined as: $f_\theta(\mathbf{x}, \mathbf{d}) \to (\mathbf{c}, \sigma)$. Later, several works [35, 15, 1, 20] have suggested extending such a design by including additional semantic outputs, enabling zero-shot segmentation capabilities. In this paper, we follow this design and extend radiance fields to produce a hierarchical feature in hyperbolic space, $h$. Fig. 1 (b) illustrates this training process.

**Loss.** Since our features live in hyperbolic space, we require a loss that properly minimizes the distance of such features. Therefore, instead of the commonly used MSE or cosine similarity, we propose to minimize the geodesic distance between the embeddings, $d_\mathbb{L}(h_i, h_j)$. Moreover, we include an additional regularization loss that encourages both embeddings to have exactly the same norm, $(d_\mathbb{L}(h_i, O) - d_\mathbb{L}(h_j, O))^2$. This additional loss helps to mitigate errors for embeddings far from $O$, where a small displacement in tangent space leads to a big displacement in hyperbolic space.

**Extrapolated features.** To supervise the hyperbolic features produced by the radiance field for a given pixel, we could select the hyperbolic feature of the object associated with it. However, since each image contains a hierarchy of objects, each pixel might be associated with multiple objects from different levels in the hierarchy. Additionally, each image contains its own hierarchy of the scene, therefore making it difficult to select a consistent feature level for supervision, which might translate to a contradictory signal during training. Therefore, we propose to extrapolate the hyperbolic features of the lowest mask in the hierarchy to the boundary of the hyperbolic space and use this extrapolated feature for supervision instead. Assuming that the sub-hierarchies present in different images are encoded in the same region of the hyperbolic space, extrapolating features will produce consistent target hyperbolic features for the supervision of the radiance field.

### 4.3 Hierarchy traversal

Once the NeRF model is trained, we can generate novel views of the scene with associated hyperbolic features per pixel. These features can then be used for open-vocabulary segmentation of objects at different granularities. Fig. 1 (c) illustrates this process of open-vocabulary query.

**Hyperbolic space traversal.** The hyperbolic space embeds the hierarchical structure of the scene, where leaf nodes are embedded close to the boundary of the hyperbolic space, and objects high in the hierarchy are embedded close to the origin. Since the NeRF has been trained to produce features $h$ on the boundary of the hyperbolic space, to traverse the scene hierarchy back to the root node, we can simply follow the geodesic path connecting the feature $h$ to the origin $O$. This continuous traversal, contrary to existing discrete approaches [15, 35, 1], allows us to dynamically sample the geodesic path with the desired number of samples $T$ at a low computational cost, resulting in a discrete number of hyperbolic features for each pixel $i$, $\{\hat{h}_{it}\}_{t=1}^{T}$. The set of features can then be decoded using the auto-encoder described in Sec. 4.1 to obtain a set of language-aligned features $\{\hat{f}_{it}\}_{t=1}^{T}$ describing the different objects of this particular branch of the hierarchy. We can compute the similarity between these features and features from a given text prompt using cosine similarity, resulting in a set of object-text similarity $\{s_{it}\}_{t=1}^{T}$.

**Aggregation method.** Previous works relying on discrete hierarchies have aggregated such similarity vectors using a max operation [15, 35] or by a fixed learned weighted sum operation [1]. While a max operation can produce good masks for simple queries, such as objects, complex compositional queries require a combination of several multi-scale responses. For such queries, in this work, we advocate for a softmax-weighted sum operation, where the list of similarity scores is normalized by a softmax operation before being used to aggregate the scores: $\phi(i) = \sum_{t=1}^{T} \hat{s}_{it} s_{it}$, where $s_{it}$ is the original similarity score for pixel $i$ at level $t$ and $\hat{s}_{it}$ is the same similarity score after being transformed by the softmax operation. This approach helps to remove noise in the set $\{s_{it}\}_{t=1}^{T}$ before aggregation, increasing high similarity scores while decreasing low ones. Unless otherwise stated, we perform all our experiments using softmax.

### 4.4 Implementation details

In order to reduce the noise in the resulting masks, following LangSplat [35] and LERF [15], we also compute the similarity of the embeddings to four neutral terms: *"object"*, *"things"*, *"stuff"*, and *"texture"*. We compute the softmax between the resulting similarity for our query and the similarity of the four neutral terms and pick the one for which the neutral softmax value is the lowest. As in LangSplat [35], the final segmentation mask is defined as the pixels with an adjusted similarity score (after softmax) higher than $0.4$.

## 5 Experiments

### 5.1 Hierarchical feature evaluation

To highlight the hierarchical understanding of our method, in the first experiment, we evaluate its ability to successfully process hierarchical open-vocabulary queries. Here, models aim to localize and segment objects and object parts in the scene, given a hierarchical query.

**Dataset.** We adapt the recent Search3D dataset [42] to the tasks of open-vocabulary segmentation and localization from radiance fields. This dataset provides instance segmentation masks at two levels of granularity: objects and object parts. The dataset provides annotations for two different sets of 3D scans, MultiScan [27] and ScanNet++ [46]. We focus on the ScanNet++ scenes, since this dataset supports novel view synthesis evaluation. In particular, we use 20 scenes from different types of rooms and select 30 unique objects and 33 object parts. As our test frames, we use the novel view synthesis split of ScanNet++.

**Hierarchical queries.** For each object-part pair, we define two prompt queries, one to identify the object and another to identify the part of the object. The object query is created by using the object class label "*[OBJ]*", which aligns with previous work on open-vocabulary segmentation. The part query is more challenging and is specifically designed to measure the hierarchical understanding of the models. This prompt is constructed by combining the labels of the object with the label of the

Table 1: Quantitative results on the Search3D [42] dataset **(a)** and in the LERF [15] dataset **(b)**. Results show that our hierarchical approach outperforms all baselines in almost all experiments.

| Method | Object IoU | Object Acc | Part IoU | Part Acc | Avg. IoU | Avg. Acc |
|---|---|---|---|---|---|---|
| LERF [15] | 25.4 | 75.0 | 06.2 | 26.6 | 15.4 | 50.0 |
| OpenNeRF [7] | 28.3 | 69.0 | 09.7 | 38.7 | 18.2 | 53.3 |
| LangSplat [35] | 41.6 | 68.9 | 11.0 | 19.3 | 25.8 | 43.3 |
| Ours | **51.4** | **89.7** | **19.9** | **50.8** | **35.1** | **69.5** |

(a)

| | Method | *ram.* | *fig.* | *tea.* | *kit.* | Avg. |
|---|---|---|---|---|---|---|
| 2D | LSeg [23] | 7.0 | 7.6 | 21.7 | 29.9 | 16.6 |
| 2D | OpenSeg [8] | 27.7 | 22.6 | 50.6 | 39.0 | 34.1 |
| 3D | LERF [15] | 28.2 | 38.6 | 45.0 | 37.9 | 37.4 |
| 3D | OpenNeRF [7] | 33.7 | 24.3 | 59.1 | 39.4 | 39.0 |
| 3D | LangSplat [35] | 51.2 | 44.7 | 65.1 | 44.5 | 51.4 |
| 3D | N2F2 [1] | **56.6** | 47.0 | 69.2 | 47.9 | 54.4 |
| 3D | Ours | 43.9 | **59.8** | **71.2** | **51.7** | **54.6** |

(b)

object part, resulting in prompts with the following form: "*[PART] of a [OBJ]*". For some pairs, we incorporate additional information in the query to differentiate the object from other objects in the scene, such as "*leg of the blue chair*".

**Metric.** We report results on the task of segmentation using Mean Intersection Over Union (mIoU) computed between the predicted mask and the annotated masks, as well as performance on the task of localization, quantified as accuracy. For localization, we follow LERF [15] and consider a true prediction if the pixel with the highest similarity score is inside the bounding box of the object.

**Baselines.** We compare our approach against state-of-the-art methods for open-vocabulary segmentation from radiance fields. We select LERF [15], OpenNerf [7], and LangSplat [35] as representative baselines. LERF and OpenNeRF exemplify NeRF-based representations, while LangSplat represents approaches based on Gaussian Splatting.

**Quantitative results.** Segmentation and localization results are reported in Tab. 1a evaluated on the adapted Search3D benchmark [42]. Results show that our method consistently achieves more accurate localization and segmentation predictions than existing approaches across all experiments. For object queries, which involve relatively simple tasks, all methods perform reasonably well in both segmentation and localization. OpenHype achieves improvements with gains of $+9.8$ in mIoU and $+14.7$ in accuracy. Composition queries, which are designed to detect object parts, pose a greater challenge as they require a hierarchical understanding of the scene. All baseline methods show a significant reduction in performance, with some achieving single-digit mIoU scores, e.g., LERF [15] obtains 6.2, and OpenNeRF [7] reaches 9.7. In contrast, OpenHype maintains robust performance, outperforming baselines by $+8.9$ in mIoU and $+12.1$ in accuracy. These results indicate that OpenHype can better capture hierarchical relationships within the scene than existing methods, effectively mitigating the *bag-of-words* effect that some baseline methods suffer from [1].

**Qualitative results.** Fig. 2 provides examples of the results of our method compared to all baselines. While most methods are able to segment objects, our method is able to provide cleaner segmentation masks with fewer artifacts. However, when more complex queries describing object parts are evaluated, all methods fail to provide an adequate mask and segment the full object instead. This is an indication that existing methods suffer from the *"bag of words"* problem. Our method, on the other hand, is able to properly detect individual object parts.

**Feature visualization.** Unlike our baselines, which encode open-vocabulary features in Euclidean space, we leverage a hyperbolic latent space. To gain insights into its structure, we provide visualizations of the hyperbolic latent space. We use CO-SNE [10] to reduce the dimension of the features to two and therefore be able to visualize them as images. Fig. 3 presents such a visualization for different images of the same scene. First, we can observe that features from different SAM masks that are part of the same hierarchy are embedded within the same geodesic path. Moreover, when we color-code the points with the level of the mask hierarchy they belong to, we can see that higher levels are embedded closer to the origin, while lower levels are embedded closer to the boundary of the hyperbolic space. Lastly, we can see that the features of masks of the same object from different views are also embedded in the same geodesic path. These visualizations provide empirical validation

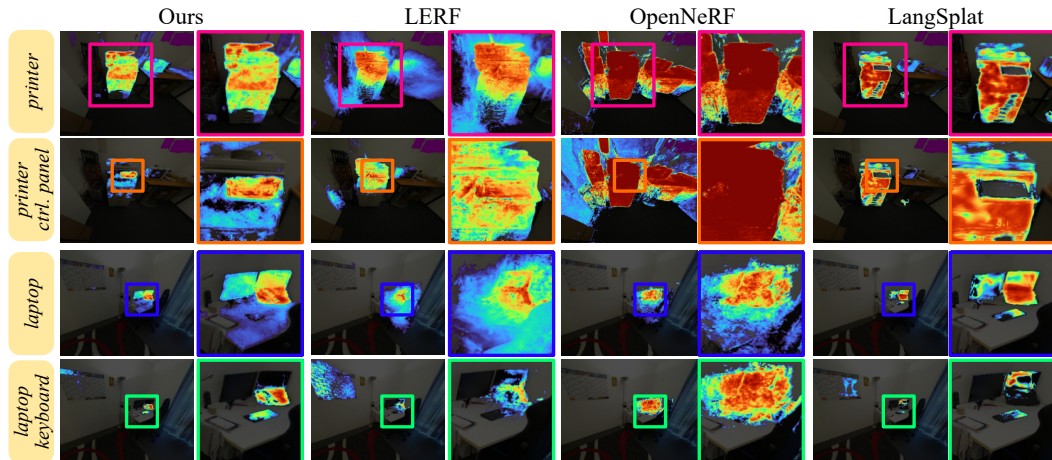

Figure 2: **Qualitative results on Search3D dataset.** While other methods struggle to segment object parts, our method successfully segments these complex compositional prompts.

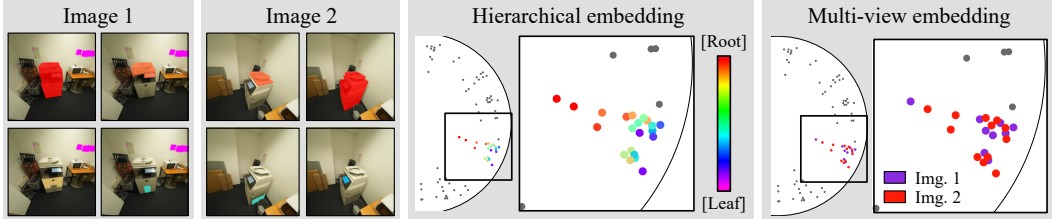

Figure 3: **Visualization of hierarchies embedded on hyperbolic features.** We visualize the hyperbolic embeddings of the hierarchies of two different images of the same scene using CO-SNE [10]. By color-coding the embeddings, we can see that hierarchies are properly embedded in the hyperbolic space even if the embeddings come from two different images of the same scene.

that our method is able to embed the mask features following the hierarchical structure defined by our masks.

## 5.2 LERF dataset

In addition to the hierarchical evaluation using Search3D, we provide open-vocabulary segmentation and localization experiments on the well-established LERF dataset [15]. This evaluation provides further insights into our improved open-vocabulary representation, aided by our hierarchical representation learning.

**Dataset.** For this task, we select the benchmark proposed by LERF [15] and later refined by LangSplat [35]. This dataset is composed of four scenes representing different environments. The mIoU of the predicted segmentation mask w.r.t. the ground truth annotated masks is reported.

**Baselines.** We reproduce the results of the same baselines as in our previous experiment, and also compare against N2F2, a Gaussian Splatting-based approach that is the most similar to our method. Since the code for N2F2 [1] was not available at the time of submission, we report results for this dataset as provided in the original paper. Moreover, we compare to the image-based approach, LSeg [23] and OpenSeg [8].

**Quantitative results.** The quantitative evaluation on the LERF dataset is reported in Tab. 1b. Results show that, even if the dataset is not focused on evaluating hierarchical capabilities of the models, OpenHype is able to provide better segmentation results than all baselines, even surpassing N2F2 [1], a recent work with discrete hierarchical representations.

Table 2: Ablation study of different components of OpenHype on the task of segmentation on five representative scenes from the Search3D dataset. Rows with the grey background ▨ indicate the configuration used in our final method.

|  | Overall | Obj. | Part |
|---|---|---|---|
| CLIP | 10.9 | 16.4 | 6.4 |
| Hyp. Tan. | 24.7 | 44.2 | 8.7 |
| Hyperbolic | 26.4 | 46.6 | 9.9 |
| + Extrap. | 30.9 | 50.4 | 14.9 |

(a) **Nerf supervision.**

|  | Overall | Obj. | Part |
|---|---|---|---|
| No Neg. | 28.7 | 45.5 | 14.9 |
| Aggregated | 30.7 | 50.4 | 14.5 |
| Stepwise | 30.9 | 50.4 | 14.9 |

(b) **Negative prompts.**

|  | Overall | Obj. | Part |
|---|---|---|---|
| Dist | 28.8 | 50.6 | 10.1 |
| Angle | NaN | NaN | NaN |
| Dist + Angle | 30.9 | 50.4 | 14.9 |

(c) **AE Loss.**

|  | Overall | Obj. | Part |
|---|---|---|---|
| Max | 29.8 | 52.5 | 11.3 |
| Mean | 30.3 | 48.8 | 15.2 |
| Sum | 24.3 | 40.0 | 11.3 |
| Softmax | 30.9 | 50.4 | 14.9 |

(d) **Path aggregation.**

|  | Overall | Obj. | Part |
|---|---|---|---|
| 3 | 19.6 | 30.9 | 10.3 |
| 16 | 30.3 | 49.2 | 14.8 |
| 32 | 30.9 | 50.4 | 14.9 |
| 64 | 28.4 | 47.0 | 13.3 |

(e) **Feature dimension.**

|  | Overall | Obj. | Part |
|---|---|---|---|
| 10 | 30.7 | 50.9 | 14.4 |
| 20 | 30.9 | 50.4 | 14.9 |
| 30 | 30.8 | 50.5 | 14.7 |

(f) **Number of steps.**

## 5.3 Ablation study

We conduct comprehensive ablation studies to systematically analyze the influence of individual components within our framework, highlighting the effectiveness of our design choices. For the evaluation, we use five scenes (showing different room types and objects) from Search3D.

**NeRF supervision.** We begin by evaluating the impact of different supervision types used during NeRF training, specifically the pixel-level features employed for distillation. We compare a model trained using CLIP embeddings from the final mask in the hierarchy of each image (CLIP) with a model trained using hyperbolic embeddings projected into the tangent space (Hyp. Tan.). Both models are supervised using an MSE objective function. Moreover, we evaluate a model trained with hyperbolic embeddings directly in hyperbolic space using our hyperbolic loss (Hyp.). Lastly, we report results for our full model, which is also trained in hyperbolic space but includes feature extrapolation toward the boundary of the hyperbolic space (Hyp. + Extrap.). Tab. 2a summarizes the results of this experiment. Hyperbolic embeddings outperform standard CLIP features. Training directly in hyperbolic space further improves performance compared to training in the tangent space. The best results are achieved with our extrapolated features, highlighting the importance of effectively addressing multi-view feature inconsistencies for achieving strong performance.

**Negative prompts.** We investigate the impact of using negative prompts on final model performance. We compare the results of the same model under three conditions: without negative prompts, with negative prompts combined after aggregation, and with negative prompts integrated at each step of the geodesic path. Specifically, in the first approach, we compute similarity scores separately for the positive and negative prompts along the geodesic path, then combine the results after path aggregation. In the last approach, we combine the similarities of positive and negative prompts at each step along the geodesic path, before aggregation. Tab. 2b shows that incorporating negative prompts improves model performance by reducing noise and variation in the similarity maps. Integrating positive and negative prompts at each step yields slightly better results than combining them after geodesic path aggregation, suggesting more effective alignment in the inference process.

**Auto-encoder loss.** We evaluate the contribution of each component in the loss function used to train our auto-encoder. Specifically, we compare models trained with contrastive learning based on the geodesic distance between masks, the exterior angle, and a combination of both. Tab. 2c shows that using only the geodesic distance yields high performance. In contrast, training with the exterior angle alone leads to a collapse of the network, indicating that it is insufficient as a standalone supervision signal. The best performance is achieved when both losses are combined, showing that they provide complementary information for learning robust representations.

**Path aggregation.** We also evaluate different variants of the aggregation method used to combine similarities along the geodesic path. Specifically, we compare selecting the maximum cosine

similarity, as done in LERF [15] (Max), summing all similarities (Sum), and computing their average (Mean). Moreover, we compare these methods to our proposed approach: a weighted sum based on softmax-normalized scores. Tab. 2d shows that while averaging (Mean) can yield competitive results, our weighted sum consistently achieves the best performance, highlighting the benefit of adaptively weighting contributions along the geodesic path. For simpler prompts, as the Object prompts, we observed that Max produces slightly better performance than Softmax.

**Feature dimension.** We investigate the impact of the hyperbolic feature dimension. We train four auto-encoders with feature dimensions of 3 (as used in LangSplat [35]), 16, 32, and 64. Tab. 2e shows that, while our method remains relatively robust across different feature dimensions, a dimension of 32 yields the best performance. In contrast, a low-dimensional embedding such as 3, as used in LangSplat, appears insufficient for capturing the hierarchical structure present in complex scenes.

**Interpolation steps.** Lastly, we evaluate the effect of the number of steps used during geodesic path traversal on model performance. Specifically, we test our model with 10, 20, and 30 steps. Tab. 2f shows that our method demonstrates strong robustness to the number of traversal steps, maintaining similar performance across all configurations.

# 6   Conclusions

In this paper, we introduce OpenHype, a novel framework for open-vocabulary segmentation of NeRF that leverages the geometry of hyperbolic space to model the hierarchical structure of 3D scenes in a continuous and flexible manner. By modeling hierarchies through geodesic traversal in hyperbolic space, OpenHype supports reasoning across multiple levels of abstraction without requiring multiple rendering passes or rigid, discrete resolutions. Our extensive evaluations across standard benchmarks demonstrate that OpenHype consistently outperforms existing methods in both quantitative metrics and qualitative analyses. These results highlight the value of incorporating continuous hierarchical priors for semantic understanding in 3D NeRF-based representations. OpenHype is agnostic to the underlying radiance field representation and is equally applicable to Gaussian Splatting. Future work could explore how our hyperbolic space can be adapted to this representation.

**Acknowledgements**

We acknowledge the EuroHPC Joint Undertaking for awarding this project (EHPC-DEV-2024D10-029) access to the EuroHPC supercomputer LEONARDO, hosted by CINECA, and the HPC system Vega at the Institute of Information Science, through an EuroHPC Development Access call.

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

# A Theoretical background

In this section, we provide additional information and a formal introduction of hyperbolic geometry concepts used in this work.

**Lorentz model.** The Lorentz model $\mathbb{L}^n$ is the upper half of a two-sheeted Hyperboloid in $\mathbb{R}^{n+1}$. More formally, this model is the $n$-dim "unit sphere" in Minkowski space defined as

$$\mathbb{L}^n := \{\boldsymbol{x} \in \mathbb{R}^{n+1} : \langle \boldsymbol{x}, \boldsymbol{x} \rangle_{\mathbb{L}} = -1/c, x_0 > 0\}, \tag{A.1}$$

where $c > 0$ denotes the curvature magnitude and $\langle \boldsymbol{x}, \boldsymbol{x} \rangle_{\mathbb{L}} = -x_0 y_0 + \sum_{i=1}^{n} x_i y_i$ is the *Lorentzian inner product*. The *Lorentzian norm* is defined as $||\boldsymbol{x}||_{\mathbb{L}} = \sqrt{|\langle \boldsymbol{x}, \boldsymbol{x} \rangle_{\mathbb{L}}|}$. The *Lorentzian geodesic distance* is defined as $d_{\mathbb{L}}(\boldsymbol{x}, \boldsymbol{y}) = \sqrt{1/c} \cdot \text{arccosh}(-c\langle \boldsymbol{x}, \boldsymbol{y} \rangle)$.

**Exponential and logarithmic maps.** The tangent space at a point $\boldsymbol{x} \in \mathbb{L}^n$ consists of Euclidean vectors orthogonal to $\boldsymbol{x}$ regarding the *Lorentzian inner product*. The *exponential map* allows us to map vectors from the tangent space of $\boldsymbol{x}$, $\mathcal{T}_{\boldsymbol{x}}\mathbb{L}^n$, to the Hyperboloid $\mathbb{L}^n$. It is defined as

$$\exp\_map_{\boldsymbol{x}}(\boldsymbol{z}) = \cosh(\sqrt{c}||\boldsymbol{z}||_{\mathbb{L}})\boldsymbol{x} + \frac{\sinh(\sqrt{c}||\boldsymbol{z}||_{\mathbb{L}})}{\sqrt{c}||\boldsymbol{z}||_{\mathbb{L}}}\boldsymbol{z}. \tag{A.2}$$

Its inverse, the logarithmic map, maps $\boldsymbol{y} = \exp\_map_{\boldsymbol{x}}(\boldsymbol{z})$, on the Hyperboloid, back to $\boldsymbol{z} \in \mathcal{T}_{\boldsymbol{x}}\mathbb{L}^n$ in the tangent space. It is defined as

$$\log\_map_{\boldsymbol{x}}(\boldsymbol{y}) = \frac{\text{arccosh}(-c\langle \boldsymbol{x}, \boldsymbol{y} \rangle_{\mathbb{L}})}{\sqrt{(c\langle \boldsymbol{x}, \boldsymbol{y} \rangle_{\mathbb{L}})^2 - 1}}(\boldsymbol{y} + c\boldsymbol{x}\langle \boldsymbol{x}, \boldsymbol{y} \rangle_{\mathbb{L}}). \tag{A.3}$$

**Traveling along geodesics.** We use geodesic paths to traverse up and down our embedded hierarchy. For this, we can use interpolation (and extrapolation) in the Lorentz model. Let $\boldsymbol{x}, \boldsymbol{y} \in \mathbb{L}^n$ be points on the Hyperboloid, $\boldsymbol{z} \in \mathcal{T}_x\mathbb{L}^n$ a tangent vector of the tangent space of $\boldsymbol{x}$ pointing in the direction of $\boldsymbol{y}$ then we can move along the geodesic starting at $\boldsymbol{x}$ going in the direction of $\boldsymbol{y}$ using

$$\gamma(t) = \cosh(t\sqrt{c}||\boldsymbol{z}||_{\mathbb{L}})\boldsymbol{x} + \sinh(t\sqrt{c}||\boldsymbol{z}||_{\mathbb{L}})\frac{\boldsymbol{z}}{\sqrt{c}||\boldsymbol{z}||_{\mathbb{L}}}, t \in [0, 1]. \tag{A.4}$$

**Exterior angle.** The exterior angle $\alpha$ used for part of our hierarchical loss $\mathcal{L}_a$ is defined as $\alpha = \pi - \angle O\boldsymbol{xy}$, where $\boldsymbol{x}$ denotes the parent of $\boldsymbol{y}$. More specifically, as introduced by the authors of [6],

$$\alpha = \arccos\left(\frac{y_0 + x_0 c\langle \boldsymbol{x}, \boldsymbol{y} \rangle_{\mathbb{L}}}{||[x_1, \ldots, x_n]||\sqrt{(c\langle \boldsymbol{x}, \boldsymbol{y} \rangle_{\mathbb{L}})^2 - 1}}\right) \tag{A.5}$$

As mentioned in section 3, we only consider the tangent space of the Hyperboloid's origin $O = [\boldsymbol{0}, \sqrt{1/c}]$. Feature vectors from the last encoder layer $\boldsymbol{z}_{enc}$ can be interpreted using $\boldsymbol{z} = [\boldsymbol{z}_{enc}, 0] \in \mathbb{R}^{n+1}$ as elements of the surrounding Minkowski space. Since $\langle O, \boldsymbol{z} \rangle_{\mathbb{L}} = 0$, those vectors are orthogonal to $O$ and hence naturally lie in the tangent space of the origin $O$ [6]. We further fix the curvature of the Hyperboloid to $-c = -1$, being a standard choice for working with hyperbolic spaces. Note that only considering the tangent space of the origin $O$ significantly simplifies the formulas for implementation as shown in [6].

# B Limitations

Our work is not without limitations. First, the quality and completeness of the extracted hierarchy and corresponding language-aligned features depend on the performance of the underlying 2D segmentation model. If object parts are not segmented in any of the 2D views, they cannot be detected or represented in the final 3D scene hierarchy.

Second, our approach relies on pretrained vision-language models (e.g., CLIP) to align visual regions with open-vocabulary text concepts. These models, while powerful, are trained on internet-scale data and may reflect biases, fail to recognize domain-specific terms, or produce inconsistent representations

for visually similar parts. This can impact the quality of the resulting scene hierarchy features and query results.

Third, inconsistent representations for visually similar parts introduced by vision-language models, as mentioned previously, introduce noise when supervising the language field across multi-view images. While we mitigate this by supervising with extrapolated features, some inconsistencies remain. A promising direction for future work is to incorporate uncertainty estimation, allowing the model to filter supervision signals and prioritize views with higher semantic agreement.

Lastly, the fidelity of the rendered and interpolated features relies on the quality of the learned hyperbolic latent space. Our approach assumes that semantics and hierarchies are smoothly distributed in this space. While our quantitative and qualitative results demonstrate that this holds to a reasonable extent, we observe some noise along geodesic paths between rendered features and the origin $O$, which can affect semantic coherence. An interesting direction for future work is to explore hyperbolic variational autoencoders, which extend hyperbolic autoencoders with probabilistic modeling that could improve the smoothness and robustness of latent inter- and extrapolations.

## C   Broader impact

This work presents a novel method for hierarchical open-vocabulary scene understanding with Neural Radiance Fields. Potential positive societal impacts include applications in robotics, assistive technologies, and augmented and virtual reality. For instance, enhanced scene understanding could improve navigation for autonomous agents, accessibility tools for individuals with visual impairments, and more intuitive human-computer interaction.

However, this technology also presents potential risks. The ability to reconstruct and label real-world scenes from images may raise privacy concerns, particularly in uncontrolled or unauthorized environments. Additionally, reliance on large-scale vision-language models introduces risks of bias propagation in scene interpretation.

We encourage responsible data collection, and fair evaluation practices. Careful deployment and user consent mechanisms will be essential to ensuring that the societal impact of this technology remains positive.

## D   Experimental details

This section provides additional details on our experimental setups and the resources necessary.

### D.1   Compute resources and efficiency

Our method does not necessitate high-performance or specialized compute resources for training or inference; most experiments were conducted on a standard GPU setup using an NVIDIA RTX A5000.

In Table D.1, the average running time on a NVIDIA RTX A5000 of the main components of our method for an image of size 543x979 with 20 interpolation steps is shown. We can observe that the most computationally expensive task is rendering, which is inherited from Nerfstudio and hence the same as for OpenNerf. Note that the rendering time also depends on the `eval_num_rays_per_chunk` variable; in our measurements, it was set to `eval_num_rays_per_chunk = 4096`. Interpolation only requires 473 milliseconds for 20 steps along the geodesic path, plus 622 milliseconds to transform hyperbolic embeddings into CLIP embeddings with our decoder. A naive discrete approach would instead require rendering of each level of the hierarchy, multiplying the rendering time by the number of levels in the hierarchy, i.e., 20 in this case. Additionally, our approach only requires a single training process, unlike methods that train multiple models.

Regarding memory usage, our method does not introduce additional overhead when sampling one level in the hierarchy. However, memory consumption increases with the number of interpolation steps, generating $n$ steps feature fields, particularly when decoding features in high-dim. CLIP space. That said, decoding can be performed in batches to manage memory constraints effectively.

Overall, our method remains substantially more efficient than training multiple models, even in memory-limited scenarios.

Table D.1: Time measurements in milliseconds for the main inference steps.

| Task | Time (ms) |
|---|---|
| rendering | 1992.214 |
| interpolating | 473.471 |
| decoding | 622.364 |

## D.2 Hyperbolic auto-encoder

**Architecture.** We use a symmetric auto-encoder with a five-layer encoder and decoder. The feature dimensions for encoder and decoder per layer are $[512, 256, 128, 64, 32]$ and $[32, 64, 128, 256, 512]$, respectively, and the activation function used is GELU [12].

**Training details.** All auto-encoders are trained for 1000 epochs using AdamW [26] as optimizer with a weight decay value of $1^{-4}$ and OneCycleLr [41] as learning rate scheduler. The initial and final division factors are 10 and 1000, the percentage of the cycle (in number of steps) spent increasing the learning rate is $5\%$. We use a batch size of 10 images, where all object features, $\mathcal{O}$, involved in at least one parent-child relationship are used. Since the number of object features, $|\mathcal{O}|$, varies between images, the batch size in number of object features varies for each pass (around 60 - 100 extracted masks/objects per image).

**Hyperbolic latent.** The output feature vectors of the last encoder layer can be interpreted as features on the tangent space of the Hyperboloid's origin $O$. We use the exponential map, $\exp\_map_O$, defined in Equation A.2, to project them onto the Hyperboloid before calculating the hierarchical loss parts $\mathcal{L}_d + \mathcal{L}_a$. Before decoding the latent features, the $\log\_map_O$ defined in Equation A.3 is applied to map them back to the tangent space. In theory, the hyperbolic space does not have a boundary as depicted in the schematic illustration of section 3. However, since distances grow exponentially when moving away from the origin, we create a "boundary orbit" by limiting the maximum geodesic distance to the origin to ensure numerical stability.

**Loss.** The loss consists of three parts, $\mathcal{L} = \mathcal{L}_d + \mathcal{L}_a + \mathcal{L}_r$. $\mathcal{L}_d$ and $\mathcal{L}_a$ are contrastive losses applied to the latent vectors on the Hyperboloid. As contrastive loss for an object $o_i$ in an image, we use

$$\mathcal{L}_{contrast}(o_i) = -log \frac{\exp(s(f_i, f^+)/\tau)}{\exp(s(f_i, f^+)/\tau) + \sum_{j=1}^{N\_\text{neg}_i} \exp(s(f_i, f_j^-)/\tau)}, \qquad \text{(D.1)}$$

to maximize the similarity $s(\cdot, \cdot)$ between positive, $f^+$, and minimize between negative pairs, $f_j^-$. For $\mathcal{L}_d$, the similarity is the negative geodesic distance on the Hyperboloid, $s \equiv -d_{\mathbb{L}}$, and for $\mathcal{L}_a$, the similarity is the negative exterior angle, as defined in Equation A.5, $s \equiv -\alpha$. For the temperature, we use $\tau = 0.2$. Each object $o_i$ used for the contrastive loss calculation has exactly one positive example, namely its direct parent. The number of negatives denoted as $N\_\text{neg}_i$ varies per object as it depends on the depth of the hierarchy the object is involved in. We use all objects of the same image that are not part of the object's $o_i$ direct hierarchy as negatives. In other words, we ignore the children or higher-level parents (e.g., parent of the direct parent) of object $o_i$ for its contrastive loss calculation, but we include siblings (objects on the same level that have the same parent) and their children as negatives to avoid a collapse of siblings on the same level to the same geodesic path. As final $\mathcal{L}_a$ and $\mathcal{L}_d$ losses we average over the respective loss $\mathcal{L}_{contrast}(o_i)$ of all objects.

Regarding the reconstruction loss, we normalize the input $f_i$ and reconstructed features $f_i'$ of the autoencoder before we calculate MSE between them, since retrieval is happening with cosine similarity, where the magnitude doesn't matter. Note that if there is an object without a child or a parent, its embedding is solely supervised by $\mathcal{L}_r$. The training duration is approximately. 40 minutes for one scene on the specified hardware.

### D.3 NeRF model

We implement OpenHype in Nerfstudio [43] and build upon the Nerfacto model. For our OpenHype vision-language field, we follow OpenNerf [7] and use an MLP with one hidden layer and a feature dimension of 256. We follow LERF [15] for the hashgrid representing language features. For optimizers and learning rate schedulers of proposal networks and fields of the underlying Nerfacto model, we use the same settings as LERF [15] and OpenNerf [7]. We train all models for 30000 steps (approx. 60 min. per scene).

### D.4 Language-aligned feature extraction

We apply Semantic SAM [24] using all granularity levels (the default setting) to extract segmentation masks of objects from a given image. We use a tight bounding box around the segmentation mask to create crops and set pixels outside of the segmentation mask to zero. Those crops are processed with CLIP [37], more specifically, we use the OpenClip [5, 13] ViT-B-16 model trained on the LAION-2B dataset (laion2b_s34b_b88k) to extract features. In order to provide a balance between object and context, we extract a second crop for each object by extending the bounding box by a factor of $0.1$ and process it with the vision-language model without zeroing out the pixels outside of the mask. The final feature $f_i$ for each $o_i$ is then the average of the two crop features.

### D.5 Querying OpenHype

As described in Sec. 4.3, querying OpenHype can be divided into $4$ steps: $1$) interpolation of rendered features in hyperbolic space, $2$) decoding interpolated features to CLIP space, $3$) computing relevancy scores for each decoded feature (representing different levels of granularity), and $4$) aggregating the relevancy scores to one single score per pixel.

**1)** For interpolation on the Hyperboloid, Equation A.4 is used to sample equidistant features along the geodesic from the rendered feature to the origin $O$.

**2)** The sampled features are mapped to the tangent space and processed with the decoder of the auto-encoder, resulting in a set of language-aligned features $\{f_{i_k}, k = 1 \dots n_{\text{steps}}\}$ per pixel $i$ with $n_{\text{steps}}$ being the number of interpolation steps.

**3)** The relevancy score $s_{i_k}$ for each of these features $f_{i_k}$ is computed using the formula introduced in [15]: $min_j \frac{\exp(f_{i_k} \cdot f_{prompt})}{\exp(f_{i_k} \cdot f^j_{neg\_prompt}) + \exp(f_{i_k} \cdot f_{prompt})}$, where $f^j_{neg\_prompt}$ are the vision-language embeddings of the negative prompts (also called canonical prompts) *"object"*, *"things"*, *"stuff"*, and *"texture"*.

**4)** The softmax-weighted mean aggregation of the relevancy scores $s_{i_k}$ per pixel is computed using $\sum_{k=1}^{n_{\text{steps}}} \beta_{i_k} s_{i_k}$, with $\beta_{i_k} = \exp(s_{i_k}) / \sum_{j=1}^{n_{\text{steps}}} \exp(s_{i_j})$. Here, $\beta_{i_k}$ can also be interpreted as attention weights computed as the softmax over the $s_{i_k}$ values. Our ablation results in Table 2d show that plain mean-aggregation yields similar results, especially for part queries. These findings are consistent with the intuition that for querying parts, the relevancy scores are high at part-level *and* object-level, distinguishing pixels that are included in the part segmentation mask from pixels included solely in the object segmentation mask. Maximum-aggregation, on the other hand, yields higher scores for objects as objects are easier to detect, and using the maximum mitigates noise on the geodesic path, which is included when aggregating using mean or softmax-weighted mean aggregation. The adapted LERF [15] dataset of LangSplat [35] has only a few prompts per scene, consisting mostly of objects. Since LangSplat and LERF use the maximum for scale selection, we follow their practice on this dataset for fair comparison. Moreover, we use the evaluation code of the public repository of LangSplat for comparability in all experiments.

### D.6 Dataset

To adapt the ScanNet++ subset of the Search3D dataset [42] to the tasks of open-vocabulary segmentation and localization from radiance fields, we project the 3D annotations to the test frames of the novel view synthesis split of ScanNet++. We work with undistorted images and sample a maximum of 250 train frames per scene evenly across all images. Note that while train and test frames are

strictly separated for training and evaluating our models, ground truth poses are used for camera parameters to fairly compare against baselines.

# E  Additional ablations and results

In this section, we provide further ablations and results that complement the findings in the main paper.

## E.1  Additional baselines based on Gaussian splats

Since we focus on radiance fields, our main baselines are based on radiance fields as well. Yet, we include the highest performing approach based on Gaussian splats, LangSplat [35], in the main results Table 1b and Table 1a. We give additional Gaussian splatting baselines in Table E.1. OpenHype outperforms the additional baselines on 2 out of 4 scenes and on average all other methods by a margin of 3.2 mIoU points.

Table E.1: Additional comparison to baselines based on Gaussian splatting on the LERF [15] dataset. Results show that our hierarchical approach outperforms all baselines on average and in almost all single scene results.

| Method | *ram.* | *fig.* | *tea.* | *kit.* | Avg. |
|---|---|---|---|---|---|
| LEGau. [40] | 13.8 | 27.6 | 45.2 | 23.7 | 27.6 |
| GOI [36] | 35.1 | 36.9 | 66.6 | 45.2 | 45.9 |
| OpenGau. [44] | 21.1 | **69.7** | 63.4 | 34.9 | 47.3 |
| LangSplat [35] | **51.2** | 44.7 | 65.1 | 44.5 | 51.4 |
| Ours | 43.9 | 59.8 | **71.2** | **51.7** | **54.6** |

## E.2  Variation between runs

As stated in Appendix B OpenHype is dependent on the quality of the latent space embeddings, where we experience some noise. Our results reported are the average over 5 runs. In Table E.2 we report the standard deviation over those 5 runs. We can see that the standard deviation remains low leaving a significant gap between OpenHype and other methods on the ScanNet++ dataset.

Table E.2: Standard deviation of the results presented in the main paper in Table 1b and Table 1a.

| Dataset | Exp. | IoU | Acc |
|---|---|---|---|
|  | Avg. | $\pm1.0$ | $\pm4.2$ |
| ScanNet++ | Obj. | $\pm1.2$ | $\pm0.0$ |
|  | Part | $\pm1.3$ | $\pm8.0$ |
| LERF | Avg. | $\pm0.6$ | - |

## E.3  Oracle ablation

We noticed that, in some cases, LangSplat has, qualitatively, clean-looking maps but selects the "wrong" level. To evaluate if OpenHype's superiority is mainly attributed to aggregation, we conduct an *oracle* experiment, where we pick the level that gives the highest IoU score for each prompt. The experiment is conducted on the same 5 scenes also used for ablations in Table 2. Results in Table E.3 show that the standard OpenHype experiment outperforms the *oracle* version of LangSplat showing the effectiveness of the continuous traversal of geodesics in our hyperbolic hierarchical embeddings. The *oracle* version of OpenHype depicts significant improvements suggesting that exploring alternative aggregation methods are an interesting direction for future work. Figure E.1 presents the relevancy maps for different prompts of objects and object parts at the 3 different levels of LangSplat and at 3 sampled levels of the continuous OpenHype hierarchy path. We can see that while

LangSplat produces qualitatively good relevancy maps even for parts, results in Table E.3 suggest that the capacity of its hard-coded 3 level hierarchy is limited. Further, Figure E.1 gives insights in the working of OpenHype's hierarchy. For the prompt "laptop keyboard" on fine granularity levels (further away form the origin $O$) the standard keyboard as well as the laptop keyboard is highlighted; when moving up along the hierarchy the relevancy map starts focusing on the laptop keyboard only. For the prompt "printer" on fine granularity levels it highlights only lightly parts, which can be clearly associated with "printer" such as the scanner-top. When moving closer to the origin $O$ the relevancy map gets stronger as the prompt corresponds more with object-levels than part-levels.

Table E.3: Results for the *oracle* experiment using 5 scenes of the ScanNet++ subset of Search3D [42]. The level that gives the highest IoU score is chosen for evaluation per prompt.

| Method | Object | Part | Avg. |
|---|---|---|---|
| LangSplat [35] | 38.3 | 5.2 | 20.1 |
| LangSplat [35] *Oracle* | 41.3 | 10.4 | 24.3 |
| Ours | 50.4 | 14.9 | 30.9 |
| Ours *Oracle* | 68.7 | 23.7 | 44.0 |

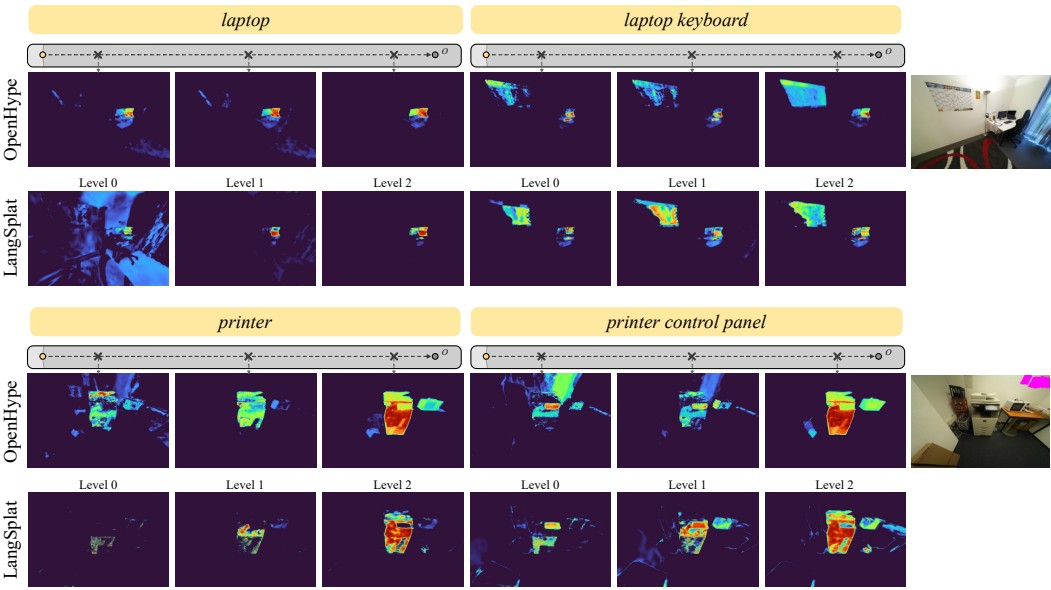

Figure E.1: Qualitative results on Search3D dataset of relevancy maps for different levels of the hierarchy of LangSplat [35] and OpenHype.

### E.4 Different foundation models

OpenHype depends on a pretrained 2D vision-language model and a pretrained image segmentation model. In the following, we investigate the impact of the model choice on OpenHype's performance using the 5 scenes of the ScanNet++ subset of Search3D [42] used in all ablation experiments.

**Vision-language model.** We use a relatively small CLIP [13, 5] model (ViT-B/16) for efficiency, and our approach still outperforms others that use the same (LangSplat, LERF) or larger base models (OpenNerf). To test the impact of a more powerful vision-language model, we conducted an ablation study using SigLIP [48], more specifically, ViT-SO400M-14-SigLIP. The results shown in Table E.4a demonstrate that our method benefits from newer models, further widening its performance advantage over state-of-the-art approaches.

At first glance, the question of why not use a pre-trained hyperbolic CLIP model might arise. A pre-trained hyperbolic CLIP encoder, such as MERU [6], is trained to encode text and images

following a visual-semantic hierarchy, where more general concepts are closer to the origin. For example, for the semantic hierarchy *"Taj Mahal" → "monument" → "architecture"*, "architecture" is the closest to the origin. In contrast to that, we use the AE to train a latent embedding that encodes the scene-specific hierarchy. This means that we can leverage the geodesic path from a rendered pixel feature to the origin in hyperbolic space, which equals traversing a vision-language feature hierarchy that encodes parts of objects or objects in the scene that the pixel is part of. The semantic hierarchies of MERU do not necessarily encode object-part relationships. Nevertheless, we have conducted an ablation using the MERU model, instead of our AE, to see if it additionally stores some object part semantic relationships that could be helpful for hierarchical scene understanding. However, as shown in Table E.4a, we observed poor performance, confirming that encoding our custom hyperbolic embedding space is key.

**Segmentation model.**    In our experiments, we use the default configuration of Semantic SAM [24], which provides six granularity prompt levels and is specifically designed for multi-level mask generation. Table E.4b presents a comparison with the native SAM [17], where we built hierarchies by combining masks from all three available levels. As shown, Semantic SAM enables the construction of more meaningful hierarchies. When comparing our results based on native SAM to LangSplat - which also leverages multiple mask levels from the native SAM - it becomes evident that, while OpenHype benefits from the richer multi-granularity masks of Semantic SAM, it consistently outperforms existing baselines across the evaluated metrics. This demonstrates that our approach is not only robust to the choice of segmentation model but also able to fully exploit finer mask hierarchies to produce superior scene representations.

Table E.4: Ablation study of different foundation models used for OpenHype on the task of segmentation on five representative scenes from the Search3D dataset.

| | Object | Part | Avg. |
|---|---|---|---|
| MERU [6] | 26.0 | 6.7 | 15.4 |
| CLIP ViT-B/16 [37] | 50.4 | 14.9 | 30.9 |
| SigLIP ViT-SO400M-14 [48] | 52.7 | 21.6 | 35.6 |

(a) **Vision-language model.**

| | Object | Part | Avg. |
|---|---|---|---|
| LangSplat [35] | 38.3 | 5.2 | 20.1 |
| native SAM [17] | 48.0 | 10.8 | 27.6 |
| Semantic-SAM [24] | 50.4 | 14.9 | 30.9 |

(b) **Segmentation model.**

## E.5    Comparison with RelationField

Recently, RelationField [20] has been introduced for modeling semantic relationships. It employs a querying strategy that separately processes nouns and relationships, subsequently combining activations. This method assumes that a prompt consists of explicitly stated relationships, such as "on top of," which makes it not compatible with the Search3D benchmark.

Yet, we provide a qualitative comparison with the "is part of" relationship querying of RelationField, which comes closest to hierarchical scene understanding. The white dot in Figure E.2 indicates the pixel of interest, whose corresponding object and object parts we aim to identify. The results of RelationField show the similarity map of its feature field with the query "is part of" with respect to this pixel of interest.

For our method, we examine the lowest-level feature of the pixel (located near a boundary) and traverse up its hierarchy by following the geodesic path to the origin of the hyperbolic space. The highlighted areas in the image correspond to pixels whose features lie within or close to the *entailment cone*, as introduced in [6], of each parent feature of our pixel of interest when moving up the hierarchy. The magnitude of the heatmap is given by the negative entailment loss [6], which equals zero when the embedding vector lies entirely within the entailment cone.

Figure E.2 shows that while RelationField highlights only part of the fridge and handles only single-level hierarchies, OpenHype naturally supports hierarchical grouping by leveraging the structure of our hyperbolic embeddings, without requiring explicit queries.

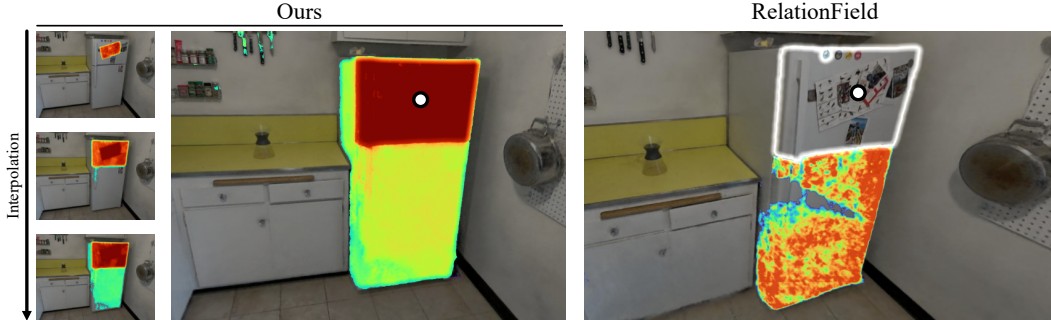

Figure E.2: **Comparison of OpenHype and RelationField [20] for the pixel of interest (white dot).** RelationField highlights only part-level relations via the "is part of" query, whereas OpenHype uncovers hierarchical groupings by traversing the geodesic path in hyperbolic space, guided by entailment cones [6].

## E.6   2D vs 3D

To demonstrate that lifting to 3D is beneficial, we conducted an ablation that uses only Semantic SAM with the extracted hierarchy and mask features directly on the evaluation frames. We compare this to the full OpenHype pipeline. Although the 2D-only variant can still segment object parts thanks to the mask hierarchy, incorporating multi-view supervision and lifting to 3D yields a clear and substantial performance improvement as shown in Table E.5.

Table E.5: Comparison between applying the Semantic SAM hierarchy and its mask features directly to the evaluation frames, versus lifting them to 3D and decoding the rendered features with the hyperbolic autoencoder.

|  | Object | Part | Avg. |
|---|---|---|---|
| Semantic SAM hierarchy [24] + CLIP [37] | 39.2 | 9.3 | 22.7 |
| Ours | 50.4 | 14.9 | 30.9 |

## E.7   Robustness study of noisy hierarchies

Since the feature field learns from multiple views ($\sim$ 250 per scene), it is able to present robust features even if some views contain noisy hierarchies. In order to evaluate it, we provide a robustness study using 5 representative scenes used in all ablations. Segmentation noise essentially results in noisy features used to supervise our model. To analyze the robustness of our model to the number of noisy views, we added Gaussian noise to the features of a number of training frames sampled randomly for training our hyperbolic auto-encoder. We use 0.2 and 0.4 standard deviation as noise level, which would result in a cosine similarity of $\sim$ 0.22 and $\sim$ 0.13, a significant drop when measuring the similarity of a normalized vector and the same vector but noised.

Results show that our model is quite robust to noisy supervision. As seen in Table E.6a, the Gaussian noise with $\sigma = 0.2$ even works as data augmentation, given slightly improved results up to 80 corrupted images. When exceeding more than half of the images to be corrupted, performance starts to decrease, yet not catastrophically.

## E.8   Results on outdoor scene

Our main results are reported on indoor scene datasets. To enable comparison with baselines in a different setting, we also annotated the Truck scene from the Tanks and Temples dataset [18]. This scene presents several notable challenges, including reflections in the vehicle windows and a large depth range with multiple elements in view. We annotated "*truck*", "*truck door*", "*wheel*", "*mudflap*", "*front bumper*", "*windshield*" as object queries and "*window of truck door*", "*logo on door*", "*wooden rails of truck bed*", "*wheel rim*", "*truck door handle*", "*truck bed*", "*truck cabin*" as part queries.

Table E.6: Robustness study results for different amount of corrupted images using Gaussian noise with different values as standard deviation.

| # corr. img. | Object | Part | Avg. |
|---:|---:|---:|---:|
| 0 | 50.4 | 14.9 | 30.9 |
| 10 | 49.5 | 15.0 | 30.5 |
| 40 | 52.8 | 15.5 | 32.3 |
| 80 | 51.3 | 14.9 | 31.3 |
| 160 | 42.6 | 14.6 | 27.2 |

(a) $\mu = 0, \sigma = 0.2$.

| # corr. img. | Object | Part | Avg. |
|---:|---:|---:|---:|
| 0 | 50.4 | 14.9 | 30.9 |
| 10 | 49.1 | 14.4 | 30.0 |
| 40 | 48.6 | 14.4 | 29.8 |
| 80 | 46.9 | 14.6 | 29.2 |
| 160 | 47.1 | 14.8 | 30.1 |

(b) $\mu = 0, \sigma = 0.4$.

Table E.7 shows that OpenHype also outperforms our baselines for part as well as object annotations on this outdoor scene. Notably, the biggest difference of more than 8 IoU points can be seen for the part-level annotations.

Table E.7: Results on the Truck scene of the Tanks and Temples dataset on the segmentation task using IoU as metric.

| | Object | Part | Avg. |
|---|---:|---:|---:|
| OpenNerf [7] | 24.98 | 18.74 | 21.7 |
| LangSplat [35] | 35.82 | 27.34 | 31.36 |
| Ours | 38.41 | 35.76 | 37.01 |

# F  List of assets

We use publicly available datasets and repositories and credit the authors properly in the paper. A full list of assets used in this work, including the license, is given below.

- Nerfstudio and its Nerfacto model [43] v1.1.5 (`https://github.com/nerfstudio-project/nerfstudio`): Apache License 2.0
- OpenClip [5, 13] v2.29.0 (`https://github.com/mlfoundations/open_clip`): MIT License
- ScanNet++ subset of Search3D [42, 46] (`https://github.com/aycatakmaz/search3d/tree/main/search3d/benchmark/docs/scannetpp_data_search3d`): released under the original ScanNet++ data terms of use
- Semantic SAM [24] (`https://github.com/UX-Decoder/Semantic-SAM`)
- SAM [17] (`https://github.com/facebookresearch/segment-anything`): Apache License, Version 2.0, January 2004
- MERU [6] (`https://github.com/facebookresearch/meru`: Attribution-NonCommercial 4.0 International
- OpenNerf [7] (`https://github.com/opennerf/opennerf/tree/main`): MIT License
- LERF [15] (`https://github.com/kerrj/lerf`): MIT License
- LangSplat [35] (`https://github.com/minghanqin/LangSplat/tree/main`): custom, research-only license created by Inria and the Max Planck Institute for Informatik (MPII)
- OpenSeg [8] (`https://github.com/donnyyou/openseg.pytorch/tree/master`): Apache License 2.0
- CO-SNE [10] (`https://github.com/yunhuiguo/CO-SNE`)

