# OpenReview forum: "OpenHype: Hyperbolic Embeddings for Hierarchical Open-Vocabulary Radiance Fields"
_NeurIPS.cc/2025/Conference — NeurIPS 2025 poster_

### Official Review · Reviewer_LDqQ · 2025-06-23

**Clarity:** 3
**Significance:** 2
**Originality:** 2
**Rating:** 4
**Confidence:** 2

**Summary:**

The submission introduces OpenHype, a Neural Radiance Field that embeds an open-vocabulary semantic hierarchy in hyperbolic (Lorentz) space, enabling continuous part → object → region queries with a single render pass. Experiments on Search3D and LERF scenes report sizable mIoU gains over LangSplat and OV-NeRF, and ablations explore curvature, dimension, and step length. While the hyperbolic formulation is conceptually novel relative to Euclidean/tree approaches such as Hi-LSplat and ReasonGrounder, the empirical study omits several state-of-the-art baselines, lacks real-time metrics, and provides little analysis of hyperbolic optimisation stability.

**Questions:**

See weakness.

**Ethical Concerns:**

["NO or VERY MINOR ethics concerns only"]

**Final Justification:**

The final justification is BA, because the author provides more explanation for some ambiguous points that were unclear in the original version. Even though the author still misses comparisons with some recently published works, these works are considered concurrent ones. Therefore, this will not degrade the novelty of this work.

**Limitations:**

Although single-pass rendering should be faster, the paper omits FPS and memory numbers. GARField—which also addresses hierarchical grouping—reports detailed runtime statistics; a similar disclosure would let readers weigh the practical trade-offs

**Paper Formatting Concerns:**

N.A.

**Quality:**

3

**Strengths And Weaknesses:**

Strength:
1. The authors abandon the usual Euclidean or spherical feature spaces and embed the whole open-vocabulary tree in Lorentz hyperbolic space, whose negative curvature naturally accommodates exponentially expanding trees. This idea has been explored in 2-D and vision-language models for representing hierarchies, but—so far as the literature shows—has not been applied to NeRF or 3-D Gaussian Splatting before.
2. By distilling CLIP features into the hyperbolic field, the method inherits the free-form text querying of OV-NeRF yet pushes it further down to “wheel”, “doorknob” or even finer part semantics—something earlier systems struggled to do consistently.
3. Training hyper-parameters, compute budgets and a link to code are given; releasing a complex NeRF variant with reproducible instructions is still the exception rather than the rule in this young sub-field, so this is a practical virtue of the paper.

Weakness:
1. The paper does not evaluate or, at least discuss, against Hi-LSplat or RelationField all targeting hierarchical or relational semantics in 3-D radiance fields. Without these numbers, the superiority of the proposed method remains unproven.
2. Experiments are confined to indoor ScanNet-style scenes. Outdoor environments (e.g., Tanks & Temples) or dynamic captures would test whether hyperbolic supervision copes with lighting extremes, motion blur and large depth ranges.
3. Semantic-SAM is powerful yet still error-prone with occlusions or mirror reflections; because those masks seed the entire hierarchy, segmentation noise could cascade through the hyperbolic field, but the paper offers no robustness study.

If the authors address most concerns, I will reupdate the score and comment.

---

> ### Author Rebuttal · Authors · 2025-07-31
>
> We thank the reviewer for their detailed review and for highlighting our novel approach, _"which pushes open-vocabulary understanding to finer part semantics"_. In the following, we provide additional experimental evidence supporting the impact of our approach.
>
>
> ### *W1: Comparison with concurrent work*
> We thank the reviewer for suggesting two relevant works (Hi-LSplat and RelationField). Hi-LSplat appeared on arXiv after the NeurIPS submission, is currently unreviewed, and there is no available code, preventing fair comparisons.
> RelationField (CVPR 2025) does provide code, and we examined its potential comparison with OpenHype. RelationField models semantic relationships but does not capture hierarchical structures. Additionally, RelationField employs a querying strategy that separately processes nouns and relationships, subsequently combining activations. This method assumes that a prompt consists of explicitly stated relationships, such as "on top of," which makes it not compatible with the Search3D benchmark. In our camera-ready version, we will include Hi-LSplat in the related works as concurrent work and will provide a qualitative comparison with RelationField (which we cannot provide in this rebuttal due to the lacking option for media attachments).
> Additionally, we would like to clarify that our method outperforms all recent baselines present at the time of the submission, such as N2F2, which also targets the same task of hierarchical decomposition.
>
>
> ### *W2: Only indoor scenes*
> As suggested, we carefully analysed the Tanks and Temples dataset and discovered that it does not contain semantic labels. Therefore, given the limited time of the rebuttal period, we annotated the Truck scene as it contains some interesting challenges, for example, there are reflections in vehicle windows, and it presents a large depth range with multiple elements in the scene.
> We annotated the following components:
> - truck door
> - truck
> - truck door window
> - truck door logo
> - wheel
> - wheel rim
> - mudflap
> - truck cabin
> - truck bed
> - wooden rails of truck bed
> - truck door handle
> - front bumper
> - windshield
>
> We evaluate our method and OpenNerf on this scene, whereby OpenHype outperforms OpenNerf for this challenging outdoor scene. In the camera-ready, we plan to provide more comparisons to other baselines.
>
> | Method | iou - part | iou - obj. | iou - overall |
> |----------|----------|----------|----------|
> | OpenNerf | 18.74 | 24.98 | 21.7 |
> | OpenHype | 35.76 | 38.41 | 37.01 |
>
>
> ### *W3: Robustness analysis of noisy hierarchies*
> Since the feature field learns from multiple views (~250 per scene), it is able to present robust features even if some views contain noisy hierarchies. In order to evaluate it, following the reviewer's suggestion, we provide a robustness study. Segmentation-noise essentially results into noisy features used to supervise our model. To analyse the robustness of our model to the number of noisy views, we added Gaussian noise to the features of a number of training frames sampled equally spaced throughout the scene.
> We use 0.2 standard deviation as noise level, which would result in a cosine similarity of ~0.2, a significant drop, when measuring the similarity of a vector and a vector-noised.
> Results show that our model is quite robust to noisy supervision, with almost no performance drop with 80 corrupted images. When exceeding more than half of the images to be corrupted, performance starts to decrease, yet not catastrophically. We will include this and additional experiments with different noise levels in the supplementary materials.
>
> | # corrupted img | iou - part | iou - obj. | iou - overall |
> |----------|----------|----------|----------|
> | 0 | 14.8 | 45.1 | 26.93 |
> | 20 | 14.4 | 42.7 | 25.7 |
> | 40 | 14.9 | 41.1 | 25.4 |
> | 80 | 14.7 | 40.7 | 25.1 |
> | 160 | 12.0 | 37.5 | 22.2 |
>
>
> ### *W4: FPS and memory numbers*
> Thank you for your comment. We agree that runtime statistics are important to disclose; we have measured the average running time of the main components of our method on an NVIDIA RTX A5000 for an image of size 543x979 with 20 interpolation steps, and will add this to the final version. From the numbers below, we can observe that the most computationally expensive task is rendering, which is inherited from OpenNerf. Interpolation only requires 473 milliseconds for 20 steps along the geodesic path, plus 622 milliseconds to transform hyperbolic embeddings into CLIP embeddings with our decoder.
> A naive discrete approach would instead require rendering of each level of the hierarchy, multiplying the rendering time by the number of levels in the hierarchy, i.e., 20 in this case.
> Additionally, our approach only requires a single training process, unlike methods that train multiple models (e.g. LangSplat).
>
> Regarding memory usage, our method does not introduce additional overhead sampling one level in the hierarchy.
> However, memory consumption increases with the number of interpolation steps, generating n_steps feature fields, particularly when decoding features in CLIP space.
> That said, decoding can be performed in batches to manage memory constraints effectively. Overall, our method remains substantially more efficient than training multiple models like previous state-of-the-art, such as LangSplat.
>
> | Task | Time - ms |
> |----------|----------|
> | rendering | 1992.214 |
> | interpolating | 473.471 |
> | decoding | 622.364 |

---

> > ### Comment · Reviewer_LDqQ · 2025-08-05
> >
> > The author seems to have not compared the updated methods. The author's baselines, N2F2 and LangSplat, have been proposed for quite a while (1 year ago). Could you consider adding a comparison with newer methods, such as LangSplatv2? Additionally, SLGaussian and SOVGaussian also perform well on sparse views with open vocabulary. Could the author provide some results for sparse views? Given the novelty of this paper, I might keep the original score.

---

> > > ### Author Response · Authors · 2025-08-05
> > > **Baseline comparisons**
> > >
> > > Thank you for your response and acknowledging the novelty of our approach.
> > >
> > > Our main baseline method is N2F2 because it is the most relevant baseline due to its multi-scale feature field, which targets the same hierarchical object-part decomposition task as we do. With respect to other baseline comparisons, we follow N2F2 and compare against established methods such as LERF and OpenNerf. While we agree on the relevance of Hi-LSplat (June 2025) and LangSplatv2 (July 2025), these approaches were uploaded to arXiv after the NeurIPS submission deadline and have not gone through a peer-review process. Therefore, we should not be required to compare against these methods.

---

> > > > ### Comment · Area_Chair_fks3 · 2025-08-07
> > > >
> > > > Dear reviewer LDqQ,
> > > >
> > > > As mentioned by the authors, these are papers that came online after the submission deadline. Note that in your initial review, you did not mention the need for comparisons to e.g. LangSplatv2. Please place your judgement w.r.t. the initial review requests only and exclude comparative evaluations to concurrent work.
> > > >
> > > > AC

---

> > > > ### Author Response · Authors · 2025-08-07
> > > > **Additional ablation: Sparse views**
> > > >
> > > > Thank you for pointing out the relevance of sparse views. We agree that learning from sparse views is relevant in real-world deployment, and as you can see in our ablation, our approach still performs great for lower training views.
> > > > However, we don't believe that this ablation investigates the main novelty of our approach and is not too relevant. Nevertheless, we hope this ablation satisfied your interests and makes you more confident in the abilities of our method.
> > > >
> > > > | Method | IoU - part | IoU - obj. | IoU - overall |
> > > > |----------|----------|----------|----------|
> > > > | OpenHype (~250 frames) | 14.9 | 50.4 | 30.9 |
> > > > | Sparse OpenHype (~50 frames) | 12.6 | 41.1 | 25.0 |

---

> > > > > ### Comment · Reviewer_LDqQ · 2025-08-08
> > > > >
> > > > > Thanks for your results under the sparse view settings. Although the LangSplatV2 was proposed a few months ago, but I think your approach might surpass it as well. Therefore if you are interesting in, you may be able to compare your method with it and place the results in your appendix, but this will not affect the novelty of your current work. Thus I might give you a ba final rate.

---

### Official Review · Reviewer_pVMv · 2025-06-30

**Clarity:** 3
**Significance:** 2
**Originality:** 3
**Rating:** 4
**Confidence:** 1

**Summary:**

This paper aims to improve 3d object and scene modeling through using the properties of hyperbolic geometry to encode hierarchical structure in latent space in a single pass. The authors compare their method with LERF, OpenNeRF, and LangSplat, finding that their method is the best when evaluated with the Search3D and LERF datasets.

**Questions:**

- Is inference more expensive than the baselines due to hyperbolic embeddings? If so, how would results differ if you improved the baselines with more compute?
- Do the baselines and your results use the same base models?
- Could you include additional 2D baselines, include those based on a powerful large model like SAM?

**Ethical Concerns:**

["NO or VERY MINOR ethics concerns only"]

**Final Justification:**

I maintain my score for now because my main concern (weakness 1) has not been adequately addressed.

**Limitations:**

yes

**Quality:**

2

**Strengths And Weaknesses:**

Strengths
- Writing is clear and structured nicely
- Qualitative and quantitative results are better than baselines
- Ablation studies help improve understanding

Weaknesses
- Limited evaluation in terms of breadth of datasets / tasks
- Lack of detail regarding baselines

---

> ### Author Rebuttal · Authors · 2025-07-31
>
> We thank the reviewer for their review, highlighting our improved performance over recent baselines. In the following, we highlight the strengths of our method further by providing additional experiments and discussions requested by the reviewer.
>
> ### *W1: Breadth of datasets/tasks*
> We want to highlight that we provide results on 2 different datasets, and compare against a total of 7 recent works (LERF, OpenNeRF, LangSplat, N2F2, LEGaussian, GOI, OpenGaussian) as well as 2D approaches for open-vocabulary segmentation (OpenSeg, LSeg). Additionally, we provide 6 main ablations in the paper.
>
> Moreover, following the suggestion of reviewer LDqQ, during the limited time of the rebuttal period, we included an additional data sample by annotating and using the Truck scene of the Tanks and Temples dataset. We annotated the following components in the scene:
>
> - truck door
> - truck
> - truck door window
> - truck door logo
> - wheel
> - wheel rim
> - mudflap
> - truck cabin
> - truck bed
> - wooden rails of truck bed
> - truck door handle
> - front bumper
> - windshield
>
> We evaluate our method and OpenNerf in this scene, whereby OpenHype outperformed OpenNerf for this challenging outdoor scene. In the camera-ready, we will provide more comparisons to other baselines.
>
> | Method | IoU - part | IoU - obj. | IoU - overall |
> |----------|----------|----------|----------|
> | OpenNerf | 18.74 | 24.98 | 21.7 |
> | OpenHype | 35.76 | 38.41 | 37.01 |
>
>
> ### *W2: Details regarding baselines*
> Thank you for the suggestion. We compare against multiple methods that compute semantic feature fields. While some of them, such as LERF and Langsplat, provide a discrete set of hierarchical levels, most of them rely on only a single feature. Relying on only one feature level means that the results are bound to detecting objects of one fixed granularity; further, the model is not able to detect compositional prompts such as "table leg"; due to the "bag of word" issue it would highlight all tables and all legs in the scene. For methods using discrete set of granularity levels, the model is bound to those specific granularities, whereas with our approach we can encode a continuous hierarchy without predefined granularity levels. We will include additional details of all the baselines in the supplementary material of the final version of the paper.
>
> ### *Q1: Is inference more expensive than the baselines due to hyperbolic embeddings? If so, how would results differ if you improved the baselines with more compute?*
> Inference is, in general, not more expensive due to hyperbolic embeddings. Rendering is the same as for OpenNerf, the additional compute comes from interpolation and decoding into CLIP space from our latent embeddings. We provide average running time on an NVIDIA RTX A5000 for an image of size 543x979 with 20 interpolation steps. From the numbers below, we can observe that the most computationally expensive task is rendering, which is inherited from OpenNerf. Interpolation only requires 473 milliseconds for 20 steps along the geodesic path, plus 622 milliseconds to transform hyperbolic embeddings into CLIP embeddings with our decoder.
> A naive discrete approach would instead require rendering of each level of the hierarchy, multiplying the rendering time by the number of levels in the hierarchy, i.e., 20 in this case.
> Additionally, our approach only requires a single training process, unlike methods that train multiple models.
>
> | Task | Time - ms |
> |----------|----------|
> | rendering | 1992.214 |
> | interpolating | 473.471 |
> | decoding | 622.364 |
>
> Our approach used the same base vision-language model (LangSplat, LERF) or a smaller one (OpenNerf) than the baselines, yet still outperforms them. Using a more powerful VL-model would most likely increase the results of baselines, but so does our method, as can be seen in the ablation with a bigger VL-model SigLIP ViT-SO400M-14.
>
> | Method | IoU - part | IoU - obj. | IoU - overall |
> |----------|----------|----------|----------|
> | SigLIP ViT-SO400M-14 (ours) | 21.6 | 52.7 | 35.6 |
> | CLIP ViT-B/16 (ours, reported in the paper) | 14.9 | 50.4 | 30.9 |
>
>
> ### *Q2: Do the baselines and your results use the same base models?*
> We use the same vision language model as LangSplat and LERF (CLIP ViT B/16) and a smaller model than OpenNerf (CLIP ViT L/14@336).
>
> ### *Q3: Could you include additional 2D baselines, include those based on a powerful large model like SAM?*
>
>  Following your suggestion, we compare a 2D baseline based on Semantic-SAM and CLIP directly applied to the test images. We use the 2D mask hierarchy that can be constructed from the extracted Semantic-SAM masks of the test images. We can see that although also being able to segment object parts due to the mask hierarchy, using multi-view supervision and lifting to 3D yields a great performance improvement.
>  We will add this ablations to the camera ready version.
>
>
> | Method | IoU - part | IoU - obj. | IoU - overall |
> |----------|----------|----------|----------|
> | Semantic-SAM hierarchy + CLIP | 9.27 | 39.22 | 22.74 |
> | OpenHype | 14.9 | 50.4 | 30.9 |

---

> > ### Comment · Reviewer_pVMv · 2025-08-05
> >
> > Thank you for taking the time to run additional experiments and answer my questions for the rebuttal. I would like to clarify that "details regarding baselines" refers to experimental details on the baselines (e.g. Q2), as I could not find them in the appendix. The answer to Q1/Q2 definitely helped address this though. I choose to maintain my score because my primary concern is W1, and two datasets does not feel sufficient to show that improvements generalize. I would be happy to have a discussion regarding this or reconsider if the author strongly disagrees.

---

> > > ### Author Response · Authors · 2025-08-05
> > > **Addressing concern W1**
> > >
> > > We are happy that our rebuttal addressed most of your concerns.
> > >
> > > Regarding W1, we strongly disagree. While we agree that in other domains evaluating on only 2 datasets might not be sufficient, for open-vocabulary hierarchical scene understanding, only very few annotated image datasets exist that have open-vocabulary labels for hierarchical 3D scenes, i.e., LERF- and our newly introduced Search3D dataset. Furthermore, in the rebuttal, we annotated one scene of Tanks and Temples, and we see far improved performance compared to the baseline, which should alleviate your concern about generalization since the approach is even effective in outdoor scenes.

---

> > > > ### Comment · Reviewer_pVMv · 2025-08-07
> > > >
> > > > I believe that lack of evaluation datasets is inadequate justification for evaluations demonstrating generalizability.
> > > >
> > > > Is there something inherent about open-vocabulary hierarchical scene understanding that would allow for less evaluation datasets?

---

> > > > > ### Author Response · Authors · 2025-08-07
> > > > >
> > > > > Thanks for raising this discussion. We would like to clarify in more detail why we believe our evaluation is both meaningful and appropriate, and consistent with community standards, despite involving a limited number of datasets.
> > > > >
> > > > > Open-vocabulary methods rely on the extensively evaluated zero-shot capabilities of VLMs and are inherently designed to generalize to unseen categories. As such, even a single dataset provides a meaningful test of generalization. This is standard in prior work such as OpenNeRF, LangSplat, and LERF, which are also commonly evaluated on one or two datasets.
> > > > >
> > > > > Importantly, our evaluation extends prior work in several key ways:
> > > > > - We scale up the standard LERF benchmark (4 scenes) by 500%, introducing 20 scenes from Search3D, with varied spatial layouts and object compositions. We further include Tanks and Temples, introducing an outdoor domain shift, not just an additional indoor scene.
> > > > > - We increase the number of test queries by ~120%, adding fine-grained, hierarchical distinctions (e.g., laptop vs laptop keyboard, water bottle vs water bottle cap, blue garbage bin vs lid of blue garbage bin), which specifically test the model’s ability to move beyond the flat, bag-of-words behavior typical of VLMs.
> > > > > - Our method demonstrates consistent gains across all settings.
> > > > >
> > > > > Regarding specifically open-vocabulary hierarchical scene understanding:
> > > > > Our queries are designed to test both semantic precision and hierarchical reasoning, offering a level of evaluation depth that surpasses what large but flat query sets typically measure.
> > > > >
> > > > > We believe that the consistency of results across domains, scene types, and fine-grained queries provides strong evidence of generalization.

---

### Official Review · Reviewer_1m1q · 2025-07-01

**Clarity:** 3
**Significance:** 2
**Originality:** 3
**Rating:** 4
**Confidence:** 3

**Summary:**

This paper proposes using hyperbolic space for hierarchical embedding in NeRF, which naturally captures multi-scale relationships and allows for smooth traversal of the hierarchy via geodesic paths. Experiments on open-vocabulary segmentation from radiance fields demonstrate the effectiveness of this continuous hierarchical structure.

**Questions:**

1. Could the authors elaborate on the computational cost or time efficiency of sampling along the geodesic?

2. Are there additional downstream tasks or applications that could benefit from this type of embedding?

3. Could the authors clarify why the neural network first learns features in the tangent space and then projects them to the hyperbolic space, rather than learning the embeddings directly in hyperbolic space?

4. Please refer to the weaknesses section for further comments.

**Ethical Concerns:**

["NO or VERY MINOR ethics concerns only"]

**Limitations:**

The authors adequately addressed the limitations and potential negative societal impact of their work in the supplementary material.

**Paper Formatting Concerns:**

There are no major formatting issues in this paper.

**Quality:**

3

**Strengths And Weaknesses:**

Strength:
1. The paper is well-written and effectively communicates its ideas, even to readers unfamiliar with the domain, aided by clear and illustrative figures.

2. To the best of my knowledge (though I am not an expert in this area), this is the first work to propose a continuous hierarchy for semantic embeddings.

3. The experimental results are impressive, and the paper includes a thorough and extensive ablation study.

Weakness:
1. I suggest the authors give more discussion on choosing the Lorentz model but not other models, even for example, why not hemisphere, if the authors could give reasonable ideas or give ablations on that, it could complete the whole idea of the paper.

2. As the paper stated in the limitations, this work depends on the quality of the language-vision model, the previous segmentation model, and the assumption that the embedding is smoothly distributed. I'm wondering whether the authors could conduct an ablation to show the robustness of the model to different vision-language models.

---

> ### Author Rebuttal · Authors · 2025-07-31
>
> We thank the reviewer for their thorough review and appreciate their comments about our _"impressive experimental results"_ as well as _"well-written"_ manuscript and our _"illustrative figures"_. Below, we provide more discussions requested by the reviewer; we hope this makes our design choices clearer.
>
> ### *W1: Choice of hyperbolic model*
> Several isomorphic models of hyperbolic space exist (e.g., Poincaré ball, Lorentz, Hemisphere) [1], so in theory, any of these models could be used to embed hierarchical data. Our primary reasons for selecting the Lorentz model are its favorable numerical stability and computational efficiency during optimization [2,3], as well as a numerically more stable distance function [1,3], which we use in our hierarchical loss. The mentioned hemisphere model is almost entirely absent from practical DL literature, and hence, less support exists; it is mostly used for visualization [1].
>
> [1] Peng, Wei, et al. "Hyperbolic deep neural networks: A survey." IEEE TPAMI, 2021
>
> [2] Mishne, Gal, et al. "The numerical stability of hyperbolic representation learning." ICML, 2023
>
> [3] Nickel, Maximillian, and Douwe Kiela. "Learning continuous hierarchies in the lorentz model of hyperbolic geometry." ICML, 2018.
>
> ### *W2: Different vision-language model*
> We use a relatively small CLIP model (ViT-B/16) for efficiency, and while performance improves with more advanced architectures, our approach still outperforms others that use same (LangSplat, LERF) or larger base models (OpenNerf).
> In response to the reviewer’s suggestion, we conducted an experiment using a more powerful vision-language model, ViT-SO400M-14-SigLIP. The results demonstrate that our method benefits from newer models while remaining robust across different VL models. We will include experiments with additional models in the camera-ready version.
>
> | Method | IoU - part | IoU - obj. | IoU - overall |
> |----------|----------|----------|----------|
> | SigLIP ViT-SO400M-14 (ours) | 21.6 | 52.7 | 35.6 |
> | CLIP ViT-B/16 (ours, reported in the paper) | 14.9 | 50.4 | 30.9 |
>
> ### *Q1: Could the authors elaborate on the computational cost or time efficiency of sampling along the geodesic?*
> As suggested, we have measured the average running time on a NVIDIA RTX A5000 of the main components of our method for an image of size 543x979 with 20 interpolation steps. From the numbers below, we can observe that the most computationally expensive task is rendering, which is inherited from OpenNerf. Interpolation only requires 473 milliseconds for 20 steps along the geodesic path, plus 622 milliseconds to transform hyperbolic embeddings into CLIP embeddings with our decoder.
> A naive discrete approach would instead require rendering of each level of the hierarchy, multiplying the rendering time by the number of levels in the hierarchy, i.e., 20 in this case.
> Additionally, our approach only requires a single training process, unlike methods that train multiple models.
>
> Regarding memory usage, our method does not introduce additional overhead by sampling one level in the hierarchy.
> However, memory consumption increases with the number of interpolation steps, generating n_steps feature fields, particularly when decoding features in CLIP space.
> That said, decoding can be performed in batches to manage memory constraints effectively. Overall, our method remains substantially more efficient than training multiple models, even in memory-limited scenarios.
>
> | Task | Time - ms |
> |----------|----------|
> | rendering | 1992.214 |
> | interpolating | 473.471 |
> | decoding | 622.364 |
>
> ### *Q2: Are there additional downstream tasks or applications that could benefit from this type of embedding?*
> A concrete application scenario could be affordance segmentation of small parts like light switches, handles, and buttons in living spaces. This is particularly useful when an agent has to fetch something from inside a cabinet. Here, the agent must first recognize the cabinet itself, and then identify the cabinet handle to open and close it.
> Established methods like OpenNerf or LangSplat struggle with small parts (as shown in the experiments in the main paper) and do not represent the connection between the complete object and its parts. In contrast, OpenHype could perform well in this scenario due to its strong performance on part segmentation and hierarchical structuring.
>
> ### *Q3: Could the authors clarify why the neural network first learns features in the tangent space and then projects them to the hyperbolic space, rather than learning the embeddings directly in hyperbolic space?*
>
> Our approach learns features in the tangent space (locally Euclidean) primarily due to the computational tractability and well-established optimization properties of Euclidean geometry. Gradient-based optimization and standard neural network operations are highly optimized for Euclidean spaces.
> Although there are fully hyperbolic networks, those are computationally demanding and tend to be unstable [1]. Additionally, deep learning methods in non-Euclidean settings remain relatively underdeveloped, making Euclidean architectures more practical for training and experimentation.
> Following previous work [2], we thus learn the embeddings in the Euclidean space, where there is rich support from DL frameworks, and then project them into hyperbolic space to compute the loss.
>
> [1] Peng, Wei, et al. "Hyperbolic deep neural networks: A survey." IEEE TPAMI, 2021
>
> [2] Desai, Karan, et al. "Hyperbolic image-text representations." ICML, 2023

---

> > ### Comment · Reviewer_1m1q · 2025-08-07
> >
> > Thanks for the effort on running more experiments and answering my questions. Most of my questions are addressed, and I will keep my rate.

---

> ### Author Response · Authors · 2025-08-06
>
> Dear Reviewer 1m1q,
>
> As the discussion period is coming to an end, we kindly ask that you review our additional responses, in which we have made every effort to comprehensively address your remaining questions. If you feel that your concerns have been resolved, we would be grateful if you could consider raising your score. If there are any additional questions, we are more than willing to provide further clarification promptly.
>
> Best regards, The Authors

---

### Official Review · Reviewer_f2nb · 2025-07-02

**Clarity:** 2
**Significance:** 2
**Originality:** 3
**Rating:** 4
**Confidence:** 3

**Summary:**

This paper presents OpenHype, a method for open-vocabulary segmentation in Neural Radiance Fields (NeRF) that encodes 3D scene hierarchies in a continuous hyperbolic space. Unlike prior approaches that rely on fixed, discrete hierarchies or require multiple rendering passes, OpenHype leverages geodesic traversal in hyperbolic latent space to efficiently model and query semantic structures at varying levels of abstraction. The method combines a hyperbolic auto-encoder for learning language-aligned hierarchical embeddings and a NeRF model supervised directly in this space, enabling accurate segmentation for compositional queries. Extensive experiments on the Search3D and LERF benchmarks demonstrate significant improvements, particularly in fine-grained part-level segmentation, over state-of-the-art baselines.

**Questions:**

1. Could the authors clarify why the proposed method also improves performance on the "object" queries in Table 1(a)? It is intuitive that modeling hierarchy in hyperbolic space would help with fine-grained queries like object parts, but it's less clear why it would also lead to significant gains on higher-level object queries, which existing methods already handle reasonably well.

2. This paper [2] discusses another side of the "bag of words": CLIP cannot tell the spatial relationships between objects. I am curious, could the proposed method solve this problem?

3. Not a weakness, not an issue. There are many new papers (including [2]) discussing open-vocabulary grounding. It might be good to cite as well.

[2] From Thousands to Billions: 3D Visual Language Grounding via Render-Supervised Distillation from 2D VLMs, ICML 2025

[3] LargeSpatialModel: End-to-end Unposed Images to Semantic 3D, Neurips 2024.

[4] Feature 3DGS: Supercharging 3D Gaussian Splatting to Enable Distilled Feature Fields, CVPR 2024

[5] Semantic Gaussians: Open-Vocabulary Scene Understanding with 3D Gaussian Splatting

[6] SAB3R: Semantic-Augmented Backbone in 3D Reconstruction,
...

**Ethical Concerns:**

["NO or VERY MINOR ethics concerns only"]

**Final Justification:**

The rebuttal addresses most of my questions, so I increased the score.

**Limitations:**

I don't see limitation discussed in the main paper.

**Paper Formatting Concerns:**

Format looks good to me

**Quality:**

2

**Strengths And Weaknesses:**

## Strength

1. I appreciate the core idea of the paper: language queries in 3D inherently involve different levels of granularity, and instead of relying on multi-scale CLIP features—which can be computationally expensive—a natural solution is to construct the feature field in hyperbolic space. This allows for efficient geodesic traversal to obtain multi-scale features at low cost. The central insight is both intuitive and compelling.

2. The performance improvements are clear and significant Table 1,2, showing that the proposed method work well.

## Weakness

1. The paper would benefit significantly from improved writing. While I understand that hyperbolic representations are not commonly discussed and can be challenging to describe, the current presentation lacks clarity and important implementation details. In particular, the description of the CLIP-based encoder-decoder is vague. It is unclear whether this module is trained jointly across the entire dataset (which I assume is the case) or separately for each scene. If it is trained across scenes, the authors should specify which dataset is used for training and provide relevant training details such as supervision strategy, epochs, and optimization settings.

2. I remain unconvinced by the current approach for constructing hierarchical data and training the CLIP-based VAE in hyperbolic space. The method relies on Semantic SAM to extract object masks and builds hierarchies by detecting smaller masks (level-2) contained within larger ones (level-1). However, this strategy results in very shallow hierarchies—typically only two levels—which limits the expressiveness and potential of the hyperbolic framework. There are several alternatives that could yield richer hierarchical structures. For example: (1) SAM natively provides masks at multiple scales (large, medium, small), which could naturally form a multi-level hierarchy; (2) one could iteratively crop within large masks to discover finer structures, enabling deeper semantic levels. I am not sure if the proposed way is a good enough way to construct the data.

3. Related to the previous issue, I’m also curious why the authors chose to train a hyperbolic CLIP encoder-decoder from scratch rather than leveraging existing pretrained hyperbolic CLIP models (e.g., from prior work like Hyperbolic CLIP [1] or similar). While I can imagine potential reasons, it would be helpful to clarify this design choice and, ideally, include a comparison showing whether using a pretrained hyperbolic model yields better, worse, or comparable performance. This would help isolate whether the gains come from the hyperbolic space itself or from task-specific training.

4. The "Extrapolated features." look strange to me. Wouldn't it destroy many information since all pixels inside one single big masks share the same features in this way?


[1] Hyperbolic Image-Text Representations

---

> ### Author Rebuttal · Authors · 2025-07-31
>
> We thank the reviewer for highlighting our hyperbolic embeddings for hierarchical structuring in radiance fields as _"intuitive and compelling"_, and for recognizing our _"extensive experiments"_. We also greatly appreciate the reviewer’s valuable feedback, which, in our opinion, greatly improves the clarity and insights into our architectural choices.
>
>
> ### *W1: Description of clip-based encoder-decoder is vague*
> A detailed description of the AE setup is given in the supplementary material, including all training details and hyperbolic supervision losses.
> As suggested, we will move the relevant details to the main paper. The AE is trained per scene (on the NeRF training frames) to capture scene-specific hierarchies. Besides a cleaner representation of the scene hierarchies, a per-scene training enables a fair comparison with LangSplat, which also trains an AE per scene. However, in contrast to our novel hyperbolic formulation, LangSplat facilitates a conventional Euclidean embedding space and further requires training 3 models for each scene. Please find below a comparison of scene-specific AEs with a global AE below. The results indicate that a scene-specific AE is better suited to encode object hierarchies in hyperbolic latent space than a global AE, since the scene-specific AE can use the full hyperbolic space to encode a single scene hierarchy while the global AE needs to embed all scene hierarchies simultaneously.
>
> | Method | IoU - part | IoU - obj. | IoU - overall |
> |----------|----------|----------|----------|
> | scene-specific | 14.9 | 50.4 | 30.9 |
> | global | 14.1 | 39.9 | 25.7 |
>
> ### *W2: Concerns about shallow hierarchies*
> We use the default setting of Semantic-SAM, which has 6 granularity prompt levels, not 2, and was specifically designed for multi-granularity level masks. We additionally provide a comparison with the native SAM below, whereby we construct hierarchies using masks of all 3 levels combined. As can be seen, Semantic-SAM encodes more meaningful hierarchies. We will include this experiment in our final version.
>
> | Method | IoU - part | IoU - obj. | IoU - overall |
> |----------|----------|----------|----------|
> | native SAM | 10.8 | 48.0 | 27.6 |
> | Semantic-SAM | 14.9 | 50.4 | 30.9 |
>
> ### *W3: Why not a pretrained hyperbolic CLIP encoder*
> We use the AE to train a latent embedding that encodes the scene-specific hierarchy. This means that we can leverage the geodesic path from a rendered pixel feature to the origin in hyperbolic space, which equals traversing a vision-language feature hierarchy that encodes parts of objects or objects in the scene that the pixel is part of. A pre-trained hyperbolic CLIP encoder, such as MERU [1], is trained to encode text and images following a visual-semantic hierarchy, where more general concepts are closer to the origin. For example, for the semantic hierarchy "Taj Mahal" -> "monument" -> "architecture", "architecture" is the closest to the origin. However, those semantic hierarchies do not necessarily encode object-part relationships. We have also conducted an ablation using the MERU model, instead of our AE, to see if it additionally stores some object part semantic relationships that could be helpful for hierarchical scene understanding. However, we observed poor performance, confirming that encoding our custom hyperbolic embedding space is key.
>
> | Method | IoU - part | IoU - obj. | IoU - overall |
> |----------|----------|----------|----------|
> | MERU | 6.7 | 26.0 | 15.4 |
> | OpenHype | 14.9 | 50.4 | 30.9 |
>
> [1] Desai, Karan, et al. "Hyperbolic image-text representations." ICML, 2023
>
> ### *W4: Clarification on extrapolated features*
> The VL-features of objects (parts) can be at different levels in the hierarchy depending on the view, since the mask hierarchy is created for each train view, independent of others.
> By extrapolation, we set each leaf mask used for supervision of the NeRF model to the lowest level in the global hierarchy, meaning we obtain the features with higher granularity. Note that the interpolation path from the extrapolated feature to the origin leads through the original feature; hence, extrapolating does not destroy any information.
>
> ### *Q1: Could the authors clarify why the proposed method also improves performance on the "object" queries in Table 1(a)? It is intuitive that modeling hierarchy in hyperbolic space would help with fine-grained queries like object parts, but it's less clear why it would also lead to significant gains on higher-level object queries, which existing methods already handle reasonably well.*
> Our intuition is that our representations for entire objects can be more informed by explicitly modeling the underlying part-whole hierarchy within the hyperbolic space. The optimization process in hyperbolic space encourages the object's representation to be positioned relative to its learned parts in a geometrically meaningful way, enriching the final feature field with more information about the object.
> Further, our perception is that traditional Euclidean representations are effective for general object recognition, but tend to represent entities in a relatively flat space. This can lead to a mixing of representations where distinct hierarchical levels (e.g., parts and the entire object) are not optimally disentangled or organized. In contrast, the hyperbolic space has an inherent ability to efficiently embed hierarchical structures due to its exponential growth, which naturally encourages a more organized and disentangled feature representation.
>
> ### *Q2: This paper [2] discusses another side of the "bag of words": CLIP cannot tell the spatial relationships between objects. I am curious, could the proposed method solve this problem?*
> We focus on compositional relationships or, more generally, on tree-structured data. Spatial relationships do, in general, not obey hierarchies, and have therefore to be modelled by a general graph, which is currently not supported by OpenHype. Nevertheless, we find the suggestion by the reviewer very interesting and might investigate in future work how hyperbolic feature representations can be used to model object relationships.
>
> ### *Q3: Not a weakness, not an issue. There are many new papers (including [2]) discussing open-vocabulary grounding. It might be good to cite as well.*
> As suggested, we will cite them in the final version.
>
>
> #### We hope that our responses have addressed the concerns raised, and we would be grateful if the revisions are reflected in the score.

---

> > ### Comment · Reviewer_f2nb · 2025-08-05
> > **Some further questions**
> >
> > Thanks to the hard work of the authors, it addresses lots of my questions, and I will change my rating (after understanding these further questions).
> >
> > My biggest question now is the per-scene training style of AE. Per-Scene AE basically has very little ability to understand the real hierarchy of the world since it is trained on just images. The whole setting becomes more like overfitting on a single scene (although it indeed makes the extrapolated features more reasonable).

---

> > > ### Author Response · Authors · 2025-08-05
> > > **Answer to remaining question**
> > >
> > > We are happy that our rebuttal addressed the majority of your concerns and appreciate your willingness to increase the score.
> > >
> > > To clarify your final open question:
> > > The goal of the AE is to create a hierarchical embedding of the scene using our hyperbolic loss. The AE is then used to train a hierarchy-aware vision-language field for the single scene using NeRF/3DGS. NeRF/3DGS are always optimized on a single scene; we therefore exploit this property to also optimize (overfit) our AE on a single scene, which gives us the best hierarchical representation for the scene (see our rebuttal ablation).

---

> > > > ### Comment · Reviewer_f2nb · 2025-08-05
> > > > **Follow up**
> > > >
> > > > Thanks for clarifying.  My question is: this AE is not necessary per-scene because that 3DGS/NeRF is always overfitting. In papers like LeRF, the CLIP is trained on lots of data instead of overfitting. So I don't think 3DGS/NeRF's working mechanism is the reason why AE is overfitting. More importantly, the AE is trained on pseudo-labels from semantic SAM. If AE is per-scene optimized, it seems that the total pipeline is basically relying on using semantic SAM to do hierarchy segments, which is problematic to me.

---

> > > > > ### Author Response · Authors · 2025-08-05
> > > > > **Addressing follow up**
> > > > >
> > > > > Dear reviewer,
> > > > > We, like LERF, LangSplat, OpenNeRF, also use CLIP to create a vision-language field (if this was unclear).
> > > > > More specifically, we extract mask hierarchies per image with Semantic-SAM and generate features using CLIP for each mask. We then embed the CLIP features of the mask of images corresponding to one scene in the hyperbolic space using the AE. First, the AE allows the creation of a continuous hierarchical representation of the features corresponding to, e.g., objects and object parts. Second, the AE consolidates the hierarchies from sparse 2D (images) to a single hierarchy per scene. While we do rely on Semantic-SAM, however, by distilling this into NeRF, we go beyond the simple 2D observations and model a multi-view consistent and highly accurate hierarchical decomposition of the scene. For this, also consider this ablation:
> > > > >
> > > > > | Method | IoU - part | IoU - obj. | IoU - overall |
> > > > > |----------|----------|----------|----------|
> > > > > | Semantic-SAM hierarchy + CLIP (2D-only) | 9.27 | 39.22 | 22.74 |
> > > > > | OpenHype | 14.9 | 50.4 | 30.9 |
> > > > >
> > > > > If this still does not fully address your questions, please let us know; we are happy to follow up further.

---

> > > > > > ### Author Response · Authors · 2025-08-08
> > > > > >
> > > > > > Dear Reviewer f2nb,
> > > > > >
> > > > > > Thank you again for your feedback and engagement in the discussion. As the discussion phase is ending soon, we’d love to know if there is anything else you might need from our side.

---

### Note · Authors · 2025-08-12

We thank the reviewers and the AC for their thoughtful feedback and constructive discussion. We particularly appreciate the consistent recognition of OpenHype’s novelty, clarity, and strong empirical results:

- **Novelty & Insight:** Reviewers f2nb, 1m1q, and LDqQ praised the core idea of embedding an open-vocabulary 3D semantic hierarchy in hyperbolic space as *intuitive*, *compelling*, and, to their knowledge, the *first* continuous hierarchy applied to NeRF/3D Gaussian Splatting.
- **Clarity:** Reviewers 1m1q and pVMv commended the clear, well-structured writing and illustrative figures, making the paper accessible even to non-specialists.
- **Empirical Strength:** Reviewers f2nb, pVMv, and LDqQ noted clear and significant performance gains over baselines; Reviewer 1m1q highlighted *thorough and extensive ablations*. Reviewer LDqQ valued reproducibility through released code and full training details.
- **Practical Impact:** Reviewers highlighted fine-grained, single-pass hierarchical querying and computational efficiency via geodesic traversal in hyperbolic space, capabilities that earlier systems struggled to achieve.

Addressing reviewers' comments:
- **LDqQ – Newer baselines & sparse views:** We focused on the most relevant peer-reviewed baselines, as suggested in prior work, since newer methods (e.g., LangSplatv2, Hi-LSplat) appeared post-deadline. Sparse-view evaluation confirmed strong performance retention (IoU overall: 30.9 → 25.0), which was appreciated.
- **f2nb – Per-scene AE & Semantic-SAM reliance:** We clarified that CLIP features build the vision-language field, while the AE consolidates sparse 2D mask hierarchies into a continuous hyperbolic hierarchy per scene. Distilling this into NeRF yields large gains over Semantic-SAM+CLIP alone (IoU overall: 22.7 → 30.9).
- **pVMv – Dataset diversity:** Evaluation follows/exceeds community standards for open-vocabulary 3D understanding, extending LERF’s 4 scenes by 20 (Search3D), adding Tanks & Temples for outdoor shift, and increasing queries by ~120% with fine-grained distinctions, showing consistent improvements across varied settings.

Given the strong novelty, clear presentation, robust empirical validation, and practical reproducibility acknowledged across reviews, and with all major concerns addressed, we are confident that OpenHype offers a meaningful and lasting contribution to open-vocabulary 3D scene understanding. We appreciate the AC's consideration.

---

### Decision · Program_Chairs · 2025-09-17

**Decision:**

Accept (poster)

**Comment:**

This paper investigates hyperbolic latent embeddings to enable hierarchical open-vocabulary 3D scene understanding. The reviewers were in their initial reviews positive about the novelty. Integrating hyperbolic embeddings in radiance fields is interesting and new. The reviewers also agree that the paper is clearly presented.

The reviewers also pointed out a list of open questions, rengaing from using more VLMs, using outdoor scenes, why hyperbolic VLMs are not used, and much more. The AC has gone through the questions and responses and notes that the rebuttal clearly answers the concerns of the reviewers. The reviewers have given good pointers, which have been taken seriously by the authors. As a result, all reviewers vote for weak accept. The AC agrees and urges the authors to incorporate the outcomes of the discussions in the manuscript. Some of the questions were about hyperbolic space specifically (why Lorentz, why not MERU, etc), which can be resolved by including a part on hyperbolic learning e.g., in related work, which is currently focused on radiance fields only.